


# Space Weather Forecasting: What We Know Now and What Are the Current and Future Challenges?

Bruce T. Tsurutani[1], Gurbax S. Lakhina[2], Rajkumar Hajra[3]

[1]Jet Propulsion Laboratory, California Institute of Technology, Pasadena, Calif, USA

[2]Indian Institute for Geomagnetism, Navi Mumbai, India

[3]National Atmospheric Research Laboratory, Gadanki, India

## ABSTRACT

Geomagnetic storms are caused by solar wind southward magnetic fields that impinge upon the Earth's magnetosphere (Dungey, 1961). How can we forecast the occurrence of these interplanetary events? We view this as the most important challenge in Space Weather. We discuss the case for magnetic clouds (MCs), interplanetary sheaths upstream of ICMEs, corotating interaction regions (CIRs) and high speed streams (HSSs). The sheath- and CIR-related magnetic storms will be difficult to predict and will require better knowledge of the slow solar wind and modeling to solve. There are challenges for forecasting the fluences and spectra of solar energetic particles. This will require better knowledge of interplanetary shock properties from the Sun to 1 AU (and beyond), the upstream slow solar wind and energetic "seed" particles. Dayside aurora, triggering of nightside substorms, and formation of new radiation belts can all be caused by shock and interplanetary ram pressure impingements onto the Earth's magnetosphere. The acceleration and loss of relativistic magnetospheric "killer" electrons and penetrating electric fields in terms of causing positive and negative ionospheric storms are currently reasonable well understood, but refinements can still be made. The forecasting of extreme events (extreme shocks, extreme solar energetic particle events, and extreme geomagnetic storms ("Carrington" events or greater)) are also discussed. Energetic particle precipitation and ozone destruction is briefly discussed. For many of the studies, the Parker Solar Probe, Solar Orbiter, Magnetospheric Multiscale Mission (MMS), Arase, and SWARM data will be useful.





# 1. INTRODUCTION

### 1.1. Some Comments on the History of Space Weather

Space Weather is a new term for an old topic. Prior to the space age where we have satellites orbiting the Earth, probing interplanetary space and viewing the Sun in UV, EUV and X-ray wavelengths, it was clearly realized that solar phenomena caused geomagnetic activity at the Earth. For example Carrington (1859) noted that there was a magnetic storm that followed ~17 hr 40 min after the well-documented optical solar flare which he reported. This storm (Chapman and Bartels, 1940) was only more recently studied in detail by Tsurutani et al. (2003) and Lakhina et al. (2012), but the hints of a causal relationship was there in 1859. Later, Hale (1931), Newton (1943) and others showed that magnetic storms were delayed by several days from intense solar flares. These types of magnetic storms are now known to be caused by interplanetary coronal mass ejections or ICMEs. Details will be discussed later in this review.

Maunder (1904) showed that geomagnetic activity often had a ~27 day recurrence, associated with some mysteriously unseen (by visible light) feature on the Sun. Chree (1913) showed that these data were statistically significant, thus inventing the Chree "superposed epoch analysis", a technique which is often used today. The mysteriously unseen solar features responsible for the geomagnetic activity were called "M-regions" by Bartels (1934) where the "M" stood for "magnetically active". It is now known that M-regions are coronal holes (Krieger et al., 1973), solar regions from which high speed solar wind streams (HSSs) emanate, causing geomagnetic activity at the Earth (Sheeley et al., 1976, 1977; Tsurutani et al. 1995). The current status of geomagnetic activity associated with HSSs and future work needed to predict the various facets of space weather events will be discussed.

With the advent of rockets and satellites, the interplanetary medium has been probed by magnetic field, plasma, and energetic particle detectors. The Sun has been viewed in many different wavelengths. The Earth's auroral regions have recently been viewed by UV imagers giving a global view of auroras including the dayside. The ionosphere has been probed by global positioning system (GPS) dual frequency radio signals, allowing a global map of the ionospheric





total electron content (TEC) in relatively high spatial and temporal resolution. The purpose of this
review article will be to give a reasonably comprehensive review of some of the major Space
Weather effects in the magnetosphere, ionosphere and atmosphere and in interplanetary space, in
order to explain what the solar and interplanetary causes are or are expected to be. The most useful
part of this review will be to focus on what future advances in Space Weather might be in the next
10 to 25 years. In particular we will mention what outstanding problems the Parker Solar Probe,
Solar Orbiter, MMS, Arase, ICON, GOLD, and SWARM data might be useful in solving.
Our discussion will first start with phenomena that occur during solar maxima (flares, CMEs and
interplanetary CME (ICME)-induced magnetic storms). We will explain to the solar scientists what
is meant by an ICME and why we distinguish this from a CME. Next, phenomena associated with
the declining phase of the solar cycle will be addressed. These include corotating interaction
regions (CIRs), high speed streams (HSSs) which cause high-intensity long-duration continuous
AE activity (HILDCAA) events, and the acceleration and loss of magnetospheric relativistic
electrons. We will then return to the topic of interplanetary shocks and their acceleration of
energetic particles in interplanetary space and also their creating new radiation belts inside the
magnetosphere. Interplanetary shock impingement onto the magnetosphere create dayside auroras
and also trigger nightside substorms. Prompt penetration electric fields during magnetic storm
main phases will be discussed with the consequences of positive and negative ionospheric storms,
depending on the local time of the observation and the phase of the magnetic storm. Two relatively
new topics, that of supersubstorms and the possibility of precipitating magnetospheric relativistic
electrons affecting atmospheric weather will be discussed. A glossary will be provided to give
definition of the terms used in this review article.
1.2. Organization of Paper
The concept of magnetic reconnection is introduced first for the nonspace plasma readers. This is
the physical process responsible for transferring solar wind energy into the magnetosphere during
magnetic storms. We have organized the rest of the paper by discussing space weather phenomena
by solar cycle intervals. However it should be mentioned that this is not totally successful since
some phenomena span all parts of the solar cycle.





Solar maximum phenomena such as Coronal Mass Ejections (CMEs), Interplanetary CMEs
(ICMEs), fast shocks, sheaths, and the forecasting of geomagnetic storms associated with the
above are covered in subsections 2.1 to 2.4. The space weather phenomena associated with the
declining phase of the solar cycle are discussed in section 3.0. Topics such as CIRs, CIR storms,
high speed solar wind streams, embedded Alfvén wave trains, High-Intensity Long-Duration
Continuous AE Activity (HILDCAA) events, relativistic magnetospheric electron acceleration and
loss, and electron precipitation and ozone depletion are discussed in subsections 3.1 to 3.6.
Although interplanetary shocks are primarily features associated with fast ICMEs and thus a solar
maximum phenomenon, shocks can also bound CIRs (~20% of the time) at 1 AU during the solar
cycle declining phase as well. Shocks and the high density plasmas that they create can input ram
energy into the magnetosphere. Topics such as solar cosmic ray particle acceleration, dayside
auroras, triggering of nightside substorms and the creation of new magnetospheric radiation belts
are covered in subsections 4.1 to 4.4. Solar flares and ionospheric total electron content (TEC)
increases is another space weather effect causing direct solar-ionospheric coupling not involving
interplanetary space nor the magnetosphere. This is briefly discussed in Section 5.0. Prompt
Penetration Electric Fields (PPEFs) and ionospheric TEC increases (and decreases) occurs during
magnetic storms. Although the biggest effects are observed during ICME magnetic storms (solar
maximum), effects have been noted in CIR magnetic storms as well. This is discussed in section
6.0. The "Carrington" magnetic storm is the most intense magnetic storm in recorded history. The
aurora associated with the storm reached 23° from the geomagnetic equator (Kimball, 1960), the
lowest in recorded history. Since this event has been used as an example for extreme space weather
and events of this type are a problem for the U.S. Homeland Security, we felt that there should be
a separate section on this topic, section 7.0. We discuss the possibility of events even larger than
the Carrington storm occurring. In section 8.0 supersubstorms are discussed. Why is this topic
covered in this paper? It is possible that supersubstorms which occur within superstorms are the
actual causes for the extreme ionospheric currents that are responsible for potential power grid
failures and not the geomagnetic storms themselves. Section 9.0 gives our summary/conclusions
for forecasting space weather events. Section 10.0 is a glossary of space weather terms used by
researchers in the field. Most of the definitions were carefully constructed and were reviewed in




a previous publication. These should be useful for an ionospheric person looking up solar terms or
vice versa.  It could be particularly useful for the nonspace plasma readership.

126                                                    W

# 2. RESULTS: Solar Maximum

## 2.1. Southward Interplanetary Magnetic Fields, Magnetic Reconnection and Magnetic Storms


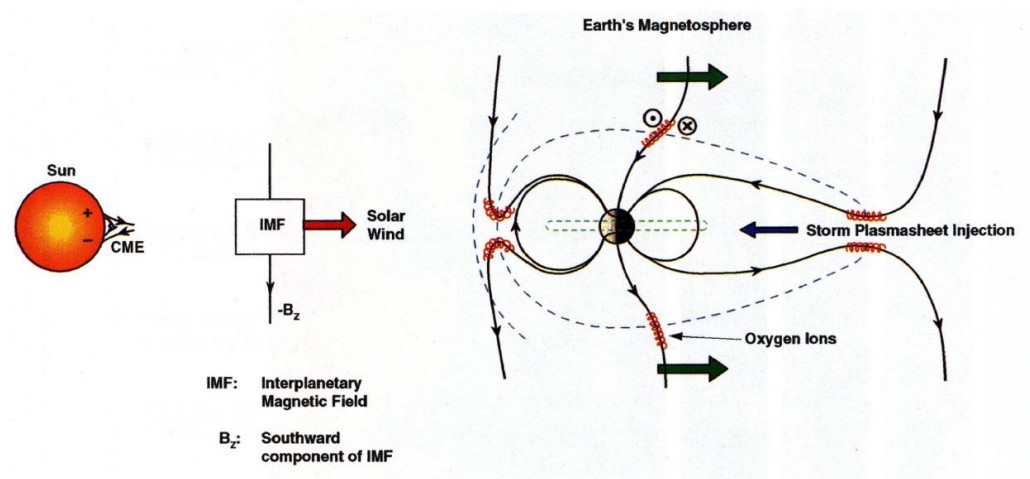


Figure 1.  Magnetic reconnection powering geomagnetic storms and substorms. Adapted from
Dungey (1961).

Figure 1 shows the Dungey (1961) scenario of magnetic reconnection. A one-to-one relationship
between southward magnetic fields and magnetic storms has been shown by Echer et al. (2008a)
for 90 magnetic storms that occurred during Solar Cycle 23.  If the interplanetary magnetic field
is directed southward, it will interconnect with the Earth's magnetopause northward magnetic
fields (the Earth's north magnetic pole is located in the southern hemisphere near the south
rotational pole). The solar wind drags the interconnected magnetic fields and plasma downstream
(in the antisunward direction). The open magnetic fields then reconnect in the tail.  Reconnection
leads to strong convection of the plasmasheet into the nightside magnetosphere.



What is known by theory and verified by observations is that the stronger the southward component of the interplanetary magnetic field, the stronger the solar wind-magnetospheric system is driven (e.g., Gonzalez et al., 1994). Intense IMF Bsouth in MCs (and sheaths) drive intense magnetic reconnection at the dayside magnetopause and intense reconnection on the nightside. Strong nightside magnetic reconnection leads to strong inward convection of the plasmasheet. The stronger the magnetotail reconnection, the stronger the inward convection. Via conservation of the first two adiabatic invariants (Alfvén, 1950), the greater the convection, the greater the energization of the radiation belt particles.

As the midnight sector plasmasheet is convected inward to lower L, the initially ~100 eV to 1 keV plasmasheet electrons and protons are adiabatically compressed (kinetically energized) so that the perpendicular (to the ambient magnetic field) energy becomes greater than the parallel energy. This leads to plasma instabilities, wave growth and wave-particle interactions (Kennel and Petschek, 1966).  The resultant effect is the "diffuse aurora" caused by the precipitation of the ~10 to 100 keV electrons and protons into the upper atmosphere/lower ionosphere. At the same time double layers are formed just above the ionosphere, giving rise to ~1 to 10 keV electron acceleration and precipitation in the formation of "discrete auroras" (Carlson et al., 1998).

 After the southward field decreases or changes orientation to northward fields, the magnetic storm recovers. The recovery is associated with a multitude of physical processes associated with the loss of the energetic ring current particles: charge exchange, Coulomb collisions, wave-particle interactions and convection out the dayside magnetopause (Kozyra et al. 1997, 2006a; Jordanova et al., 1998; Daglis et al. 1999).  A typical time for storm recovery is 10 to 24 hrs (Burton et al., 1975; Hamilton et al., 1988; Ebihara and Ejiri, 1998; O'Brien and McPherron, 2000; Dasso et al., 2002; Kozyra et al., 2002; Wang et al., 2003; Weygand and McPherron, 2006; Monreal MacMahon and Llop, 2008).

**2.2. Coronal Mass Ejections (CMEs), Interplanetary Coronal Mass Ejections (ICMEs) and Magnetic Storms**

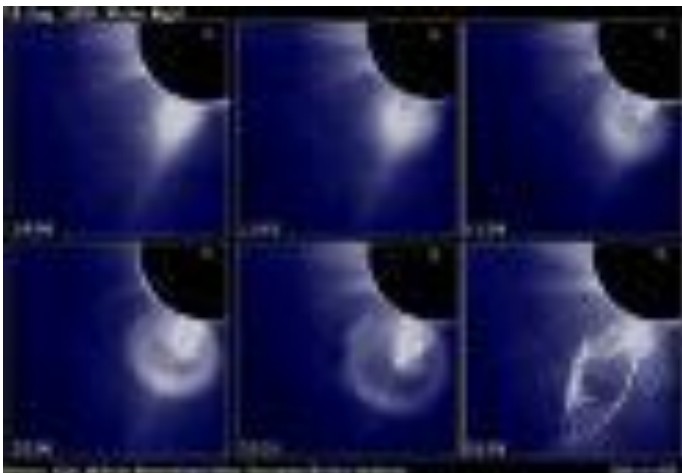

174

Figure 2. A sequence of images showing the emergence of parts of a coronal mass ejection (CME)
coming from the Sun. The time sequence starts at the upper left and ends  at the lower right. Taken
from Illing and Hundhausen (1986).

What are the solar and interplanetary sources of intense interplanetary magnetic fields that lead to
magnetic reconnection at Earth and intense magnetic storms? What we know from space age
observations is that these magnetic fields come from parts of a coronal mass ejection, a giant blob
of plasma and magnetic fields which are released from the Sun associated with solar flares and
disappearing filaments (Tang et al., 1989).  Figure 2 shows the emergence of a CME from behind
a solar occulting disc.  The time sequence starts at the upper left, goes to the right and then to the
bottom left, and ends at the bottom right.  The three parts of a CME are best noted in the image on
the bottom left.  There is a bright outer loop most distant from the Sun, followed by a "dark region",
and then closest to the Sun is the solar filament.

**2.3**. **Forecasting Magnetic Storms and Extreme Storms Associated with ICMEs**

We will precede ourselves and state here that for the limited number of cases studied to date, the
most geoeffective part of the CME is the "dark region". Interplanetary scientists (Burlaga et al.,
1981; Choe et al., 1982; Tsurutani and Gonzalez, 1994) have identified this as the low plasma beta
region called a magnetic cloud (MC), first identified by Burlaga et al.(1981) and Klein and Burlaga
(1982) in interplanetary space by magnetic field and plasma measurements.  When there are





southward component magnetic fields within the magnetic cloud (thought to typically be a giant
fluxrope), a magnetic storm results (Gonzalez and Tsurutani, 1987; Gonzalez et al. 1994; Zhang
et al., 2007; Echer et al. 2008a).

Interplanetary and magnetospheric scientists have developed the term ICME or interplanetary
CME because it is not known how the CME evolves as it propagates from the Sun to the Earth and
beyond.  For example the bright outer loops are seldomly identified at 1 AU (one rare case was
identified by Tsurutani et al., 1998) and the filaments are typically not found within the ICME at
1 AU. A rare case was reported by Burlaga et al. (1998).  For statistical results we direct the reader
to Lepri and Zurbuchen (2010).  Where have the bright outer loops and filaments gone to? Have
they simply detached only to impinge onto the magnetosphere at a later time, or do they go back
into the Sun?  Observations from the Parker Solar Probe, Solar Orbiter and ACE plus ground-
based solar observations could perhaps help address this question.

It should be remarked that the high density solar filaments could be extremely geoeffective if they
collided with the Earth's magnetosphere (this is covered later in Section 3.2.5). Modeling and
examining the Parker Solar Probe and Solar Orbiter data could help us understand whether the MC
evolves as it propagates through interplanetary space.  Is it possible for the MC to rotate so that
initially southward magnetic fields become northward components?  Can the MC fields be
compressed or expanded by interplanetary interactions? Can magnetic reconnection be taking
place within the ICME between the solar corona and 1 AU as suggested by Manchester et al.
(2006) and Kozyra et al., (2013)?  If so, how often does this occur and can it be predicted?

Of course the most important goal for space weather is predicting the southward magnetic fields
within the ICME.  This extremely difficult task is the holy grail of space weather.  It is more
important than predicting the time of the release of a CME, its speed and its direction.

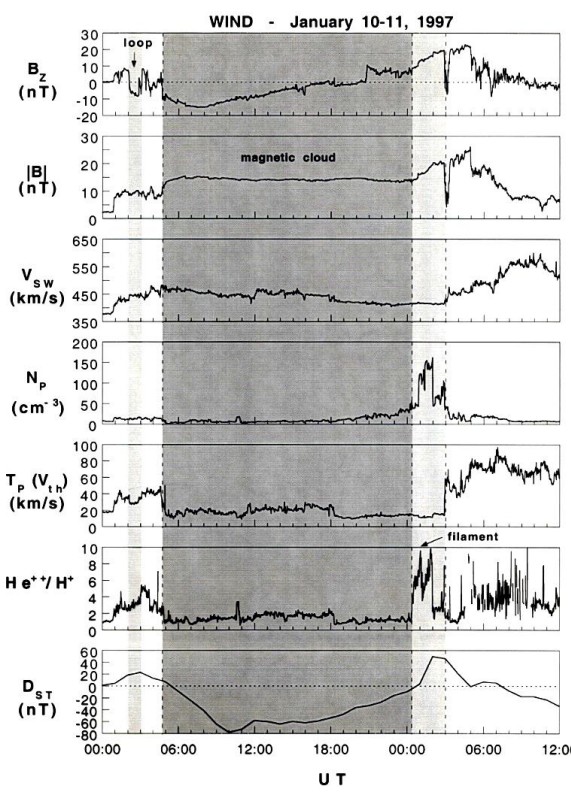


Figure 3. An ICME detected at 1 AU just upstream of the Earth.

Figure 3 shows a rare case of an ICME where all three parts of a CME are detected at 1 AU.  The
MC is indicated by the shaded region in the figure. The outer loop was identified by Tsurutani et
al. (1998) and the filament by Burlaga et al. (1998).

From top to bottom are the interplanetary magnetic field (IMF) Bz component (in geocentric solar
magnetospheric/GSM coordinates), the field magnitude, the solar wind velocity, density,
temperature and the $He^{++}/H^+$ ratio.  The bottom panel gives the ground based Dst index whose
amplitude is used as an indicator of the occurrence of a magnetic storm. Dst becomes negative
when the Earth's magnetosphere is filled with storm-time energetic ~10-300 keV electrons and
ions (Williams et al., 1990).  Dessler and Parker (1959) and Sckopke (1966) have shown that the
amount of magnetic decrease is linearly related to the total kinetic energy of the enhanced radiation
belt particles. This is because the energetic particles which comprise the storm-time ring current,





through gradient drift of the charged particles, form a diamagnetic current which decreases the
Earth's magnetic field inside the current. We refer the reader to Sugiura (1964) and Davis and
Sugiura (1966) for futher discussions of the Dst index. The Dst index is a one hr index.  More
recently a 1 min SYM-H index (Iyemori, 1990; Wanliss and Showalter, 2006) has been developed.
This is more useful for high time resolution studies. Both indices are produced by the Kyoto Data
Center.

In this example (top panel of Figure 3) the MC fields start with a strong southward (Bz< 0 nT)
component and then later turns northward.  In the bottom panel, the magnetic storm Dst index
becomes negative with very little delay from the southward magnetic fields.  The energy transfer
mechanism is magnetic reconnection (Dungey, 1961) as discussed in Section 2.1.  The high density
filament (fourth panel from the top) is present after the MC passage. Values as high as ~160 cm$^{-3}$
have been detected. These values are extreme values with the nominal solar wind density being ~
3 to 5 cm$^{-3}$ (Tsurutani et al., 2018a). The high densities impinging on the magnetosphere in this
case caused the Dst index to reach a maximum of ~+55 nT.

The stronger the southward component of the MC fields, the more intense the magnetic storm at
the Earth.  In extreme cases storms with intensities of Dst < -250 nT  can occur (Tsurutani et al.
1992a; Echer et al. 2008b).  An empirical relationship between the speed of the MC at 1 AU and
its magnetic intensity has been shown by Gonzalez et al. (1998). A hypothetical explanation is the
"melon seed model": squeezing a melon seed will cause it to squirt out, squeezing it harder will
make it come out fast. A larger magnetic field will require greater pressure to release it.  However
a real MHD or plasma kinetic model is need to explain this empirical relationship.

Because extremely strong MC magnetic fields are needed to produce extreme magnetic storms
like the "Carrington" event (Tsurutani et al., 2003; Lakhina and Tsurutani, 2017), one should focus
on extremely fast events for forecasting purposes. The geoeffective interplanetary dawn-to-dusk
electric field is Vsw x Bsouth.  Because Gonzalez et al (1998) have shown that |B| is empirically
proportional to Vsw, the dawn-to-dusk interplanetary electric field has a Vsw$^2$ dependence. The
Carrington ICME took ~17 hr 40 min to go from the Sun to Earth (Carrington, 1859) causing the
largest magnetic storm in history, Dst estimated to be -1760 nT.  However the August 1972 event





was even faster, taking only ~14 hr 40 min to go from the Sun to Earth (Vaisberg and Zastenker
1976; Zastenker et al. 1978). Although the 1972 MC was indeed extreme in speed and magnetic
field intensity, the direction of the magnetic field was northward and thus geomagnetic quiet during
the MC impingement onto the magnetosphere (Tsurutani et al. 1992b). So again, predicting the
ICME magnetic field direction is paramount in importance.

Modeling of ICME propagation in interplanetary space during disturbed AR periods has met only
limited success (Echer et al., 2009; Mostl et al., 2015). Sometimes it is difficult to even identify
which flare or disappearing filament a detected ICME is related to (see Tang et al., 1989). The
propagation times from the Sun to 1 AU has often been in error by days (Zhao and Dryer, 2014).
The additional information provided by the Parker Solar Probe and Solar Orbiter and examination
of present ICME propagation codes will help improve the ability to make more accurate forecasts.

**2.4. Fast Shocks, Sheaths and Magnetic Storms**

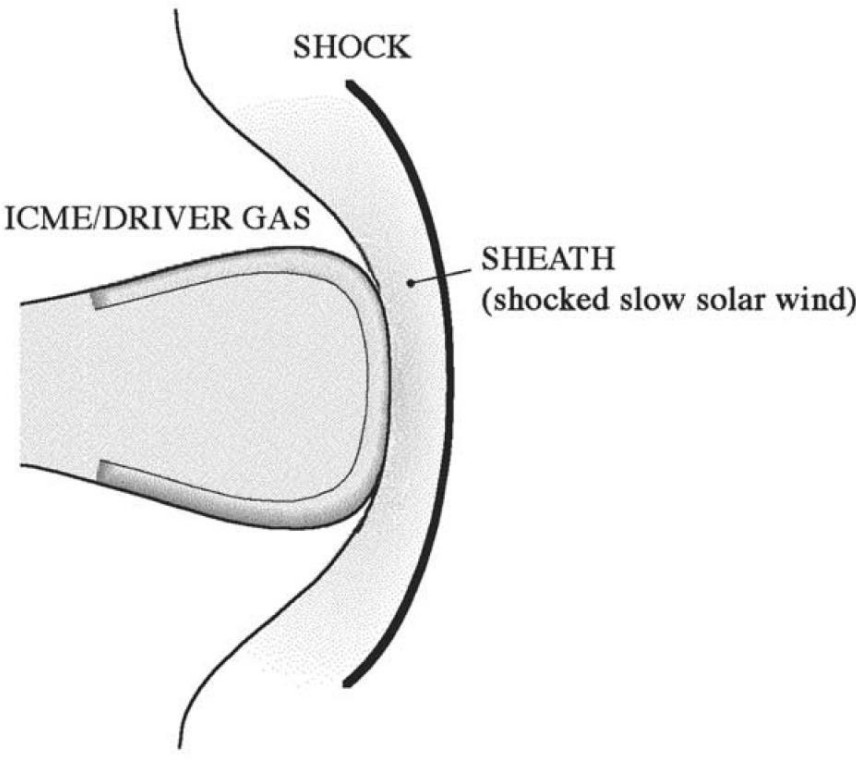


Figure 4.  A schematic of an interplanetary sheath antisunward of an ICME. In this diagram the

Sun is on the left (not shown).


Figure 4 shows a schematic of a shock and sheath upstream of an ICME. "Fast" CMEs/ICMEs can

create upstream fast forward shocks (Tsurutani et al., 1988). By "fast" it is meant that the

CME/ICME is moving at a speed higher than the upstream magnetosonic (fast wave mode) speed

relative to the upstream plasma and by "forward" we mean that the shock is propagating in the

same direction as the "driver gas" or the CME/ICME, antisunward.  When a shock is formed, it

compresses the upstream plasma and magnetic fields.  In this terminology, the upstream direction

is the direction in which the shock is propagating (antisunward in this case) and the downstream

direction is towards the Sun (see Kennel et al., 1985 and Tsurutani et al., 2011 for details on

shocks).  The compressed plasma and magnetic fields downstream of the shock is the "sheath".

The shock and sheath are not part of the CME/ICME.  The origin of this plasma and magnetic

fields is the slow solar wind, altered by shock compression. This is important to realize if one





wishes to predict magnetic storms caused by interplanetary sheath southward magnetic fields. It
should be noted that "slow" ICMEs have been detected at 1 AU (Tsurutani et al., 2004a). These
phenomena do not necessarily have upstream shocks and sheaths, as expected. However the
southward MC magnetic fields still cause magnetic storms.

Kennel et al. (1985) used MHD simulations to show that the plasma densities and magnetic field
magnitudes downstream of shocks are roughly related to the shock magnetosonic Mach numbers.
This relationship holds up to a Mach number of ~4. For higher Mach numbers MHD predicts that
the compression will remain at a factor of ~4. Since interplanetary shocks detected at 1 AU
typically have Mach numbers only of 1 to 3 (Tsurutani and Lin, 1985; Echer et al., 2011; Meng et
al. 2019), 1 to 3 are the typical shock magnetic field and density compressions detected at 1 AU.
One question for future studies is "does the MHD relationships of magnetic field magnitude and
density jumps hold for extreme shocks?" If not, there will be important consequences for extreme
space weather.

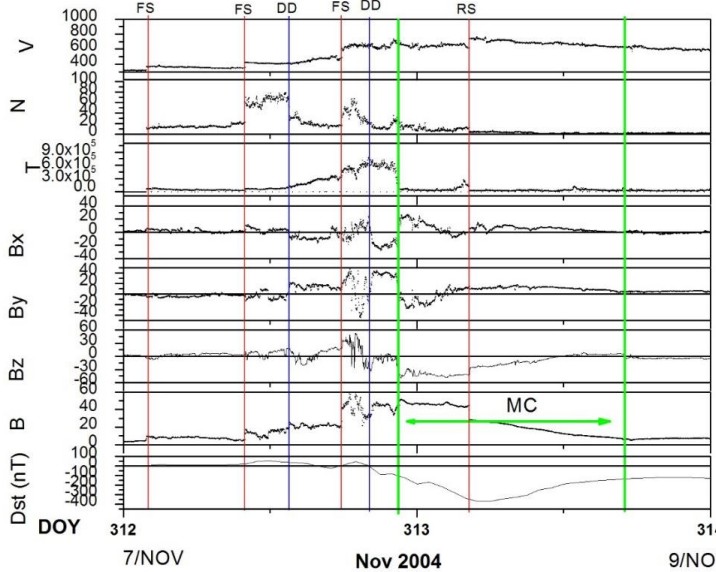


Figure 5. An example of three fast forward shocks pumping up the interplanetary magnetic field
intensity. Taken from Tsurutani et al. (2008a).

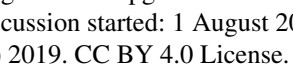




Figure 5 shows a complex interplanetary event that was selected by the CAWSES II team to study
in detail. The full information on this event from the Sun to the atmosphere can be found in the
special issue: Large Geomagnetic Storms of Solar Cycle 23
(https://agupubs.onlinelibrary.wiley.com/doi/toc/10.1002/(ISSN)1944-8007.CYCLE231). What
is important is that this event was associated with a solar active region (AR) and the results are
quite important in terms not only for interplanetary disturbance phenomena but also for
geomagnetic activity at the Earth.

From top to bottom in Figure 5 are the solar wind speed, density, and temperature, the IMF Bx,
By and Bz components and the magnetic field magnitude in GSM coordinates. In this coordinate
system, **x** points in the direction of the Sun, **y** is $(\Omega \times \mathbf{x})/|\Omega \times \mathbf{x}|$ where $\Omega$ is the Earth's south
magnetic pole and **z** completes the right hand system. The magnetic storm Dst index is given at
the bottom. Fast forward shocks are denoted by the three vertical red lines on 7 November 2004.
There are sudden increases in the velocity, density, temperature and magnetic field magnitude at
all three events. The Rankine-Hugoniot relationships have been applied to the plasma and
magnetic field data to determine that they are indeed fast shocks.

The point of showing this case is to indicate that each shock pumps up the interplanetary sheath
magnetic field by factors of ~2 to 3. The initial magnetic field magnitude started with a value of
~4 nT and at the peak value after three shocks, it reached a value of ~60 nT. This final value was
higher than the MC magnetic field which was ~45 nT. Details concerning the shocks and
compressions can be found in the original paper for readers who are interested. What is important
here is how intense interplanetary magnetic fields are created. They can come from the MCs
themselves or the sheaths, as shown here. However in this case the southward magnetic fields that
caused the magnetic storm came from the MC and not the sheath.

In the above example it is believed that three fast forward shocks were associated with three ICMEs
released from the AR. The longitudinal extent of shocks are, however, wider than the MCs, so
only one MC was detected in the event. A similar situation was found for the August 1972 event
discussed earlier.

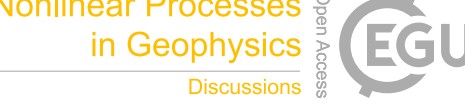


It should be noted that a fast reverse wave (here by "reverse" we mean that the wave is propagating
in the solar direction) was detected during the Figure 5 event. It is identified as the red vertical line
on 8 November. In detailed examination of the Rankine-Hugoniot conservation equations, this
wave was found to propagate at a speed below the upstream magnetosonic speed and thus was a
magnetosonic wave and not a shock. This reverse wave caused a decrease in the MC magnetic
field (and the southward component) and thus the start of the recovery phase of the magnetic storm.
The reader should note that fast reverse waves and shocks are also important for geomagnetic
activity. A detailed discussion of shock and discontinuity effects on geomagnetic activity can be
found in Tsurutani et al. (2011).

**2.4.1. Forecasting ICME sheath magnetic storms**

Determination of the IMF Bz component in the sheaths will be a difficult task. To do this, more
effort on predicting the slow solar wind plasma and magnetic field will be required. To date, there
has been little effort expended in this area. This is, however easy for us to hope for, but in practice
is far more difficult to do. Use of data from Solar Probe, Solar Orbiter and a 1 AU spacecraft such
as ACE will help in these analyses.

This problem has recently been emphasized by results from Meng et al. (2019). Meng et al. have
shown that superstorms (Dst < -250 nT) that occurred during the space age (1957 to present) are
mostly driven by sheath fields or a combination of sheath plus a following magnetic cloud (MC).

Substorms are generated by lower intensity southward magnetic fields with the process of
magnetic reconnection being the same as above. However substorm plasmasheet injections only
go in to L ~4, the outer part of the magnetosphere (Soraas et al., 2004). The auroras associated
with substorms appear in the "auroral zone", 60° to 70° magnetic latitudes (MLATs). Magnetic
storms associated with much larger IMF Bsouth are detected at subauroral zone latitudes.

## 3.0. RESULTS: Declining Phase of the Solar Cycle
**3.1. Corotating Interaction Region (CIR) Magnetic Storms**

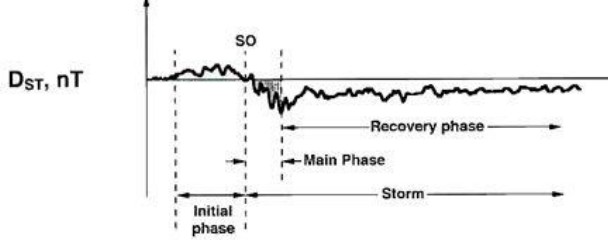


Figure 6. The magnetic Dst profiles of a CIR magnetic storm (bottom) and an ICME magnetic

storm (top). Taken from Tsurutani (2000).


During the declining phase of the solar cycle a different type of solar and interplanetary activity

dominates the cause of magnetic storms, that of solar coronal holes and Corotating Interaction

Regions (CIRs). The magnetic storms caused by CIRs are quite different from storms caused by

ICMEs and/or their sheaths. Figure 6 shows the difference in profiles of two different types of

magnetic storms. The profile of a CIR magnetic storm is shown on the bottom and that of a shock

sheath ahead of an ICME MC magnetic storm on top.

388

The ICME MC magnetic storm Dst profile, discussed briefly earlier (see Figure 3), is reasonably

easy to identify (top panel). There is a sudden, ~tens of second duration positive increase in Dst

which is caused by the sudden increase in solar wind ram pressure caused by the passage of the

sheath high density jump downstream of the shock. This compresses the magnetosphere, creating

the sudden impulse (SI[+]: see Joselyn and Tsurutani, 1990) detected everywhere on the ground

(Araki et al., 2009). Later, in either the sheath or the MC there may be a southward IMF which

causes the magnetic storm. If there is a southward component in the MC, it is usually smoothly





varying in intensity and direction.  This leads to a smooth monochromatic storm main phase as
seen in the Dst index (and illustrated in the Figure 6 (and Figure 3). The loss of the ring current
particles is the storm recovery phase. The details of storm recovery phase durations and causative
mechanisms will be an interesting topic for magnetospheric scientists to study in the near future.
The Arase mission data will be quite useful for these studies.

The bottom panel of Figure 6 shows the typical profile of a CIR magnetic storm. It is quite different
from a MC magnetic storm profile. There is no $SI^+$ associated with the beginning of the
geomagnetic disturbance.  This is because CIRs detected at 1 AU typically are not led by fast
forward shocks (Smith and Wolf, 1976; Tsurutani et al. 1995).  The positive increase in Dst is
associated with the impact of a high density region near the heliospheric current sheet (HCS)
(Smith et al., 1978; Tsurutani et al. 2006a) called the heliospheric plasmasheet (Winterhalter et al.,
1994) and/or associated with the compressed plasma at the leading edge of the CIR.  These are
slow solar wind plasma densities. The most distinguishing feature of the CIR storm main phase is
the lack of smoothness, in sharp contrast to the MC magnetic storm.  This irregular Dst storm main
phase is caused by large Bz fluctuations within the CIR.

CIR magnetic fields have magnitudes of ~20 to 30 nT and typically do not reach the much higher
magnetic field intensities that MC fields do.  For this reason and also because of the Bz
fluctuations, CIR magnetic storms are typically have intensities  Dst $\geq$ -100 nT (smaller magnetic
storms).  Extreme magnetic storms with Dst < -250 nT caused by CIRs are rare, if they occur at
all (none found in the Meng et al. 2019 study). However it is clear that compound events involving
both CIRs and ICMEs could certainly cause extreme magnetic storm events.

CIR related magnetic storms occur most frequently during the declining phase of the solar cycle
and ICME magnetic storms typically occur near the maximum phase of the solar cycle.  However
have said that, it should be noted that both CIR storms and ICME MC magnetic storms can occur
during any phase of the solar cycle. We have simply ordered things by solar cycle so that it will
be easier to give the reader the general picture of space weather.

**3.2 Coronal Holes, High Speed Solar Wind Streams and Geomagnetic Activity**





**3.2.1. Coronal holes and high speed solar wind streams**

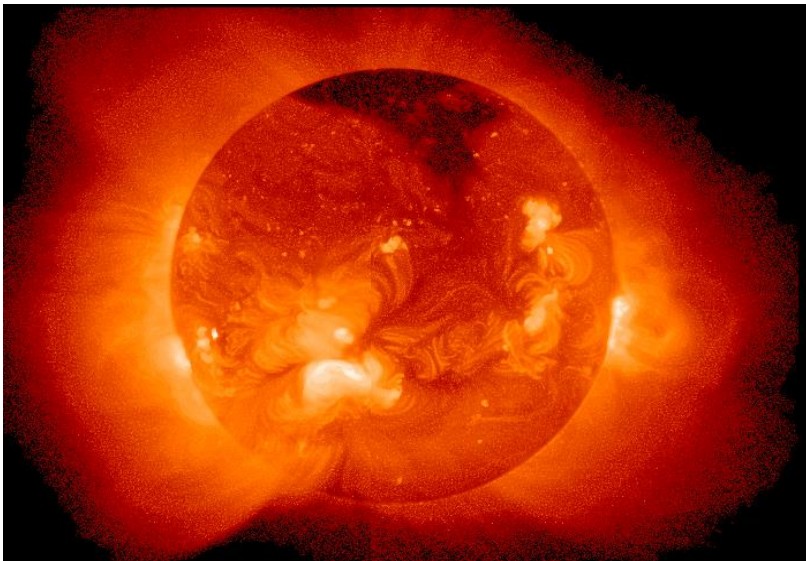


Figure 7.  A giant polar coronal hole near the north pole of the Sun.

Figure 7 shows a polar coronal hole at the north pole of the Sun. This image was taken by Solar
Dynamic Observatory, NASA (https://sdo.gsfc.nasa.gov/) in soft x-rays showing the dark (low
temperature) region at the pole.  Large polar coronal holes occur typically in the declining phase
of the solar cycle (Bravo and Otaola, 1989; Bravo and Stewart, 1997; Zhang et al., 2005).

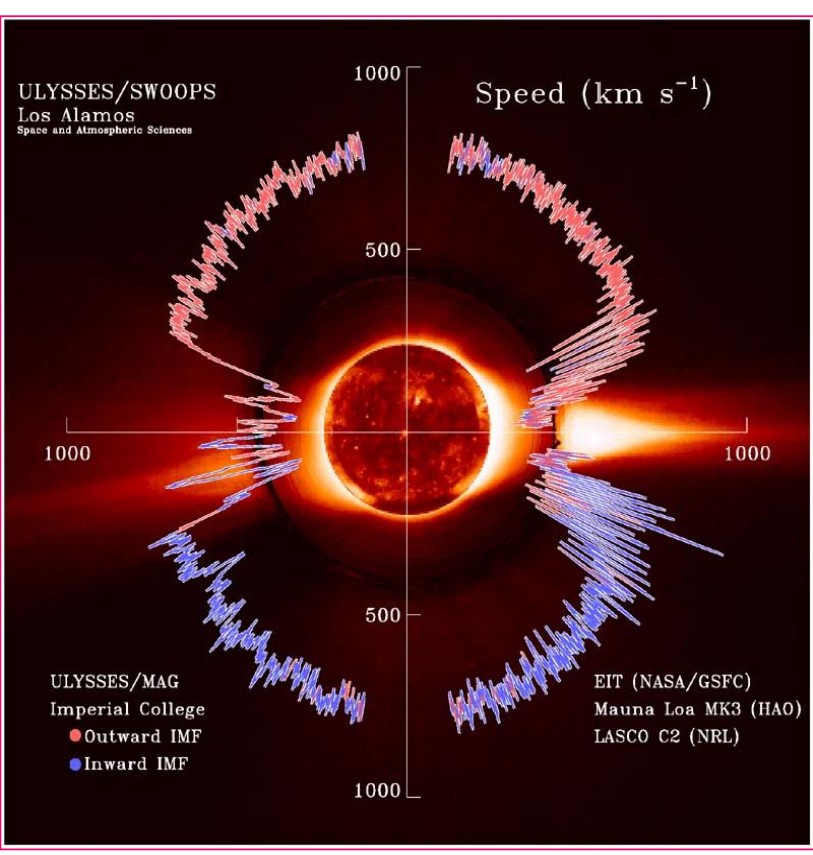


Figure 8.  High speed solar wind streams emanating from coronal holes in the north and south

solar poles. The figure was taken from Phillips et al. (1995) and McComas et al. (2002).


Figure 8 gives a "dial plot" of the solar wind speed for the first traversal of the Ulysses spacecraft

over the Sun's poles.  The radius from the center of the Sun to the trace indicates the solar wind

speed.  The magnetic field polarity is indicated by the color of the trace, red for outward IMFs and

blue for inward IMFs. A SOHO EIT soft x-ray image of the Sun is placed at the center of the figure

and a High Altitude Observatory Mauna Loa coronagraph image is superposed onto the Figure.

445

Two large polar coronal holes are detected at the Sun, one at the north pole and the other at the

south pole.  It is noted that HSSs of ~750 to 800 km/s are detected at Ulysses when over the polar

coronal hole regions. When Ulysses was near the solar equatorial region where helmet streamers

are present, the solar wind speeds are of the slow solar wind variety, Vsw ~ 400 km/s.  The reader





should note that it took years for Ulysses to make this polar orbit while the solar and coronal

images were taken at one point in time. However this composite figure is useful to illustrate the

main points about the origins of HSSs.

**3.2.2 High speed solar wind streams and the formation of CIRs**

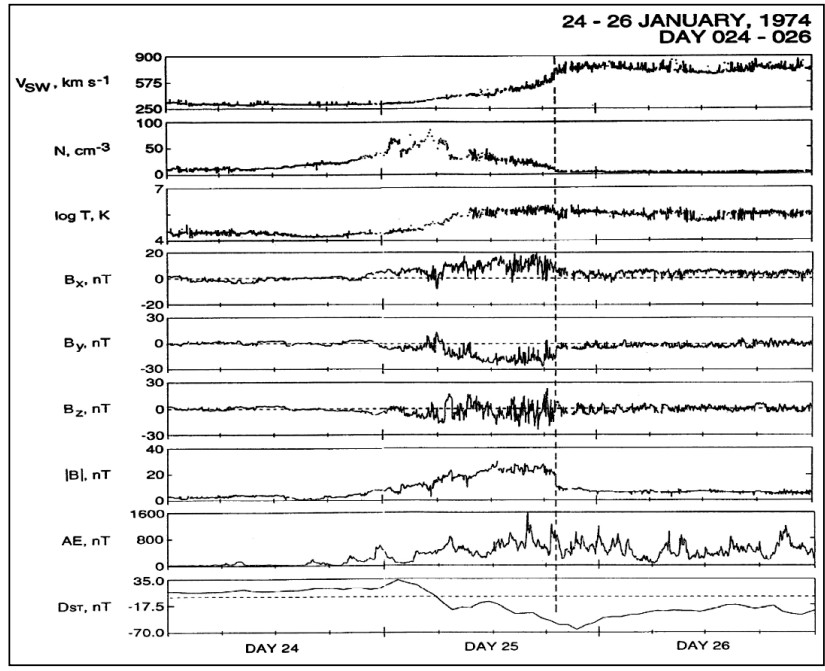

Figure 9. A high speed solar wind stream-slow solar wind interaction and the formation of a CIR.
The format is the same as in Figure 4 except that the AE index is given in the next to bottom panel.
The figure is taken from Tsurutani et al. (2006a).

Figure 9 shows a HSS-slow speed stream interaction. The right portion of the top panel on day 26

shows a HSS with speeds of 750-800 km/s at 1 AU. On day 24, the top panel left indicates a solar

wind speed of ~300 km/s, or the slow solar wind. The effects of the stream-stream interaction

occurs on day 25. This is best seen in the IMF magnitude panel, 7th from the top. The stream-

stream interaction creates intense magnetic fields of ~25 nT. The 6th from the top panel is the IMF

Bz component (in GSM coordinates). The Bz is highly fluctuating. Magnetic reconnection

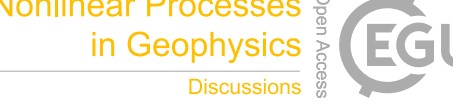

between the IMF southward components and the magnetopause magnetic fields leads to the

irregularly shaped storm main phase shown in the bottom (Dst) panel.

To be able to forecast a CIR magnetic storm, one would have to first understand the sources of the

IMF Bz fields. For example are they compressed upstream Alfvén waves (Tsurutani et al. 1995,

2006b)? Or could they be waves generated by the shock interaction with upstream waves in the

slow solar wind? That would be only the first step for forecasting, of course. Then with knowledge

of the properties of the slow speed stream, the details of the wave compression/interaction would

then have to be calculated/modeled.

Another approach would be to determine if there is an underlying southward component of the

IMF within the CIR. This would most likely be caused by the geometry of the HSS-slow speed

stream interaction and may be predictable from MHD modeling. If this is correct, then the wave

fluctuations can be modeled as being superposed on top of these dc magnetic fields. The Parker

Solar Probe, Solar Orbiter and ACE data could be useful in these endeavors.

### 3.2.3. High speed solar wind streams, Alfvén waves and HILDCAAs

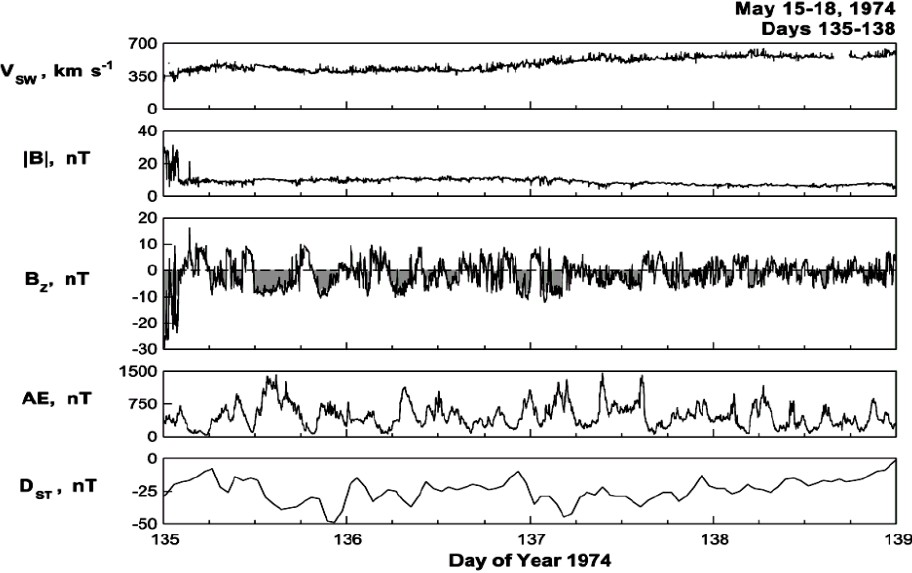





Figure 10. A high-intensity, long-duration continuous AE activity (HILDCAA) event. Taken from
Tsurutani et al. (2006b).

The schematic in Figure 6 showed a long "recovery phase" that trails the CIR magnetic storm main
phase (see Tsurutani and Gonzalez, 1987 and Yermolaev et al. 2014 for a contrast in
interpretation). However we now know that the storm wasn't "recovering" as in the case of an MC
magnetic storm but that something else was occurring. This "recovery" can last from days to
weeks. Thus processes of charge exchange, Coulomb collisions, etc. for particle losses are not
tenable to explain such long "recoveries".

Figure 10 shows the interplanetary cause of this extended geomagnetic activity. It occurs primarily
during HSSs independent of whether a CIR magnetic storm occurred prior to it or not (Tsurutani
and Gonzalez, 1987; Tsurutani et al., 1995, 2006a; Kozyra et al. 2006b; Turner et al. 2006; Hajra
et al. 2013, 2014a, 2014b, 2014c, 2017). From top to bottom are the solar wind speed, the IMF
magnitude, the IMF $B_z$ component (in GSM coordinates) and the auroral electrojet (AE) index.
The bottom panel is the Dst index.

The interplanetary data were taken from the IMP-8 spacecraft, an Earth orbiting satellite that was
located upstream of the magnetosphere in the solar wind at this time. The location was inside 40
Re, where an Re is an Earth radius. The magnetic $B_z$ fluctuations have been shown to be Alfvén
waves which are of large nonlinear amplitudes in HSSs (Belcher and Davis, 1971; Tsurutani and
Gonzalez, 1987; Tsurutani et al., 2018b). What is apparent from this figure is that every time the
IMF $B_z$ is negative (southward), there is an AE increase and a Dst decrease. This has been
interpreted as being due to magnetic reconnection between the southward components of the
Alfvén waves and the Earth magnetopause. The AE is enhanced by the same magnetic
reconnection process that occurs during substorms, and a small parcel of plasmasheet plasma is
injected into the nightside magnetosphere suppressing the Dst index slightly. It is noted that there
are many southward IMF $B_z$ dips in this four day interval of data shown in Figure 10. There are
also many corresponding AE increases and Dst decreases. Thus the interpretation of the
constant/average Dst value of ~ -25 nT for four days is that continuous plasma injection and decay
is occurring. This is clearly not a "recovery phase" where the ring current particles are simply



lost, it only appears as a recovery from the Dst trace. Soraas et al. (2004) have shown that particles
are injected during these events but only to L values of 4 and greater. These are shallow injections
as suggested above.

These geomagnetic activity events have been named High-Intensity, Long-Duration Continuous
AE events or HILDCAAs (Tsurutani and Gonzalez, 1987).  This is simply a description of the
events without an interpretation. In 2004 when a detailed examination using Polar EUV auroral
imaging was applied, it was found that many phenomena besides simple isolated substorms
occurred (Guarnieri, 2006; Guarnieri et al., 2006). Although substorms occur during HILDCAA
events, there are AE increases (injection events?) that are not well-correlated with substorm onsets
(Tsurutani et al., 2004b).  The full extent of HILCAAs is not well understood (see also Souza et
al., 2016, 2018; Mendes et al., 2017). Data from SWARM, MMS and Arase could help answer
this question.

There is also the question of the origin of the interplanetary Alfvén waves? Do they originate at
the Sun caused by supergranular circulation, or is that mechanism untenable as argued by Hollweg
(2006)? Could the waves be generated locally between the Sun and Earth as speculated by Matteini
et al. (2006, 2007) and Hellinger and Travnicek (2008)? The Parker Solar Probe and Solar Orbiter
mission data could be useful in helping answer these questions.

The original requirement for identifying a HILCAA event was quite strict.  The event had to occur
outside of a magnetic storm main phase (Dst was required to be > -50 nT: Gonzalez et al. 1994),
the peak AE intensity had to be greater than 1,000 nT (high-intensity), the event had to last longer
than 2 days (long-duration), and there could not be any dips in AE less than 200 nT for longer than
two hrs (continuous).  Clearly there are HILDCAAs with the same interplanetary causes and
geomagnetic effects as for the strict definition.  However the strict definition is useful for further
studies using different data sets.

**3.2.4. HILDCAAs and the Acceleration of Relativistic Magnetospheric Electrons**

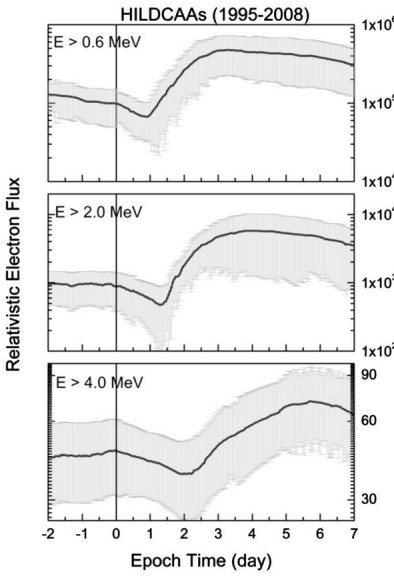


Figure 11. The relationship between HILDCAAs and relativistic electron acceleration. The figure
is from Hajra et al. (2015a).

One of the consequences of HSSs and HILDCAAs is the acceleration of relativistic electrons.
These energetic particles can damage orbiting satellite electronic components (Wrenn, 1995).
Figure 11 shows the relationship between the onset of HILCAA events (vertical line) and
relativistic electron fluxes. From top to bottom are the E > 0.6 MeV, the E > 2.0 MeV and the E
> 4.0 MeV electrons detected by the GOES-8 and GOES-12 satellites located at L = 6.6. This
figure is a superposed epoch analysis (Chree, 1913) result of all of the HILDCAA events in solar
cycle 23 which are not preceded by magnetic storms. This was done to avoid contamination by
storm-time particle acceleration. The zero epoch time (vertical line) corresponds to the HILDCAA
onset time. Here the "strict" definition of HILDCAAs was used to define the onset times.

The figure shows that the appearance of E > 0.6 MeV electrons is statistically delayed by ~1.0 day
from the onset of the HILDCAAs. The E > 4.0 MeV electrons are statistically delayed by ~2.0
days from the HILDCAA onset. It is thus possible that HILCAAs may be used to forecast
relativistic electron enhancements in the magnetosphere (see Hajra et al., 2015b; Tsurutani et al.,
2016a; Hajra and Tsurutani, 2018a; Guarnieri et al., 2018). This however has not been done yet
and could be implemented by scientists today.






The physics for the electron acceleration has been well-developed by magnetospheric scientists.
Two competing acceleration mechanisms have been developed. In one mechanism, with each
injection of plasmasheet particles on the nightside magnetosphere, the anisotropic ~10 to 100 keV
electrons generate electromagnetic whistler mode chorus waves (Tsurutani and Smith, 1974;
Meredith et al. 2002) by the loss cone/temperature anisotropy instability (Brice, 1964; Kennel and
Petschek, 1966; Tsurutani et al., 1979; Tsurutani and Lakhina, 1997).  The chorus then interacts
with the ~100 keV injected electrons to energize them to ~0.6 MeV energies (Inan et al., 1978;
Horne and Thorne, 1998; Thorne et al., 2005, 2013; Summers et al., 2007; Tsurutani et al., 2010;
Reeves et al., 2013; Boyd et al., 2014).  The lower-frequency part of the chorus in turn interact
with the ~0.6 MeV electrons to accelerate them to ~2.0 MeV energies, etc.  This bootstrapping
mechanism has been suggested by several authors (Baker et al., 1979, 1998; Li et al., 2005; Turner
and Li, 2008; Boyd et al., 2014, 2016; Reeves et al., 2016).

An alternative scenario is that relativistic electrons are created through particle radial diffusion
driven by micropulsations (Elkington et al., 1999, 2003; Hudson et al., 1999; Li et al., 2001, O'Brien et
al., 2001; Mann et al.,2004; Miyoshi et al., 2004). However the same general scenario would hold as for
chorus acceleration.  The substorms and convection events within HILDCAAs would be the sources for the
micropulsations and the micropulsations would last from days to weeks in duration.  Bootstrapping of
energy would still take place.

A few important questions for researchers are: "How high can the relativistic magnetospheric
electron energy get?".  If there are two HSSs, one from the south pole and another from the north
pole so that Earth's magnetosphere is bathed in HSSs for years, as happened during 1973-1975
(Sheeley et al., 1976, 1977; Gosling et al. 1976; Tsurutani et al. 1995), will the energies go above
~10 MeV?  What will physically limit the energy range? This is important for keeping Earth-
orbiting satellites safe during such events.

**3.2.5. Solar wind ram pressure pulses and the loss of relativistic electrons**



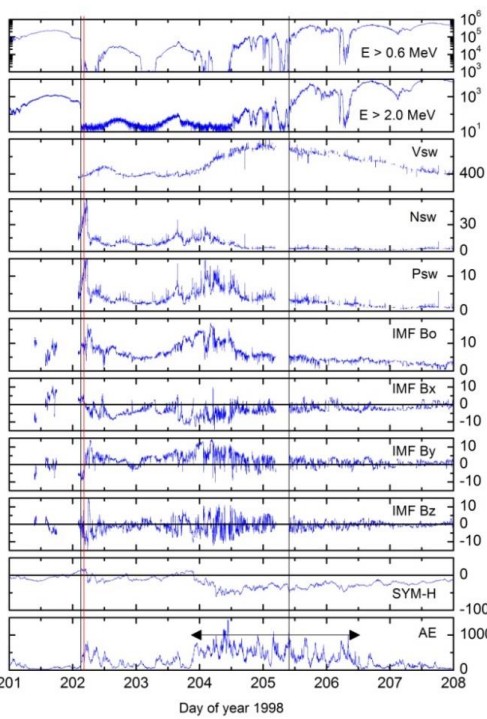


Figure 12. A relativistic electron decrease (RED) event and later acceleration. Taken from
Tsurutani et al. (2016b).


Figure 12 shows a relativistic electron decrease (RED) event. From top to bottom are the E > 0.6
MeV electron fluxes, the E > 2.0 MeV electron fluxes, the solar wind speed, density and ram
pressure, and the IMF magnitude, Bx, By and Bz component in the GSM coordinate system. The
bottom two panels are the 1 min SYM-H index (think of this as a high resolution Dst index:
Wanliss and Showalter, 2006) and the AE index. The relativistic electron measurements were
taken at L = 6.6.

604

At the beginning of day 202, a vertical black line indicates the onset of a high density heliospheric
plasmasheet (HPS: Winterhalter et al., 1994) that is identified in the fourth panel from the top. The
HPS is by definition located adjacent to the HCS (Smith et al. 1978). The HCS is noted by the
reversal in the signs of the IMF Bx and By components (seventh and eighth panels from the top).
The onset of the HPS is followed within one hr by the vertical red line, the sudden disappearance





of the E > 0.6 MeV (first panel) and E > 2.0 MeV (second panel) relativistic electrons. Tsurutani
et al. (2016b) has shown that for 8 relativistic electron disappearance events during solar cycle 23
all of the disappearances were associated with HPS impingements onto the magnetosphere.

Where have the relativistic electrons gone?  There are two primary possibilities. One is that the
energetic electrons have gradient drifted out of the magnetosphere through the dayside
magnetopause, a feature that has been called "magnetopause shadowing" by West et al. (1972).
However a second possible mechanism is electron pitch angle scattering by electromagnetic ion
cyclotron (EMIC) waves.  We think that this second possibility is more intriguing and has far more
interesting consequences, if correct. One might ask where the EMIC waves come from and why is
pitch angle scattering particularly important?  It has been shown by Remya et al. (2015) that when
the magnetosphere is compressed, both electromagnetic chorus (electron) waves (Thorne et al.,
1974; Tsurutani and Smith, 1974; Meredith et al. 2002) and EMIC (ion) waves (Cornwall, 1965;
Kennel and Petschek, 1966; Olsen and Lee, 1983; Anderson and Hamilton, 1993; Engebretson et
al., 2002; Halford et al. 2010; Usanova, 2012; Saikin, 2016) are generated.  The compression of
the magnetosphere causes betatron acceleration of remnant ~10 to 100 keV electrons and protons,
and thus plasma instabilities associated with both particle populations occur. What is particularly
important is that the EMIC waves are coherent (Remya et al., 2015), leading to extremely rapid
pitch angle scattering of ~ 1 MeV electrons by the waves. The scattering rate has been shown to
be three orders of magnitude faster than that with incoherent waves (Tsurutani et al., 2016b).

Another possible loss mechanism is associated with possible generation of PC waves by the HPS
impingement followed by radial diffusion of the relativistic electrons. Wygant et al. (1998) and
Halford et al. (2015) have mentioned that larger loss cone sizes at lower L could be a source of
loss to the ionosphere.  Rae et al. (2018) has shown that superposition of compressional PC waves
and the conservation of the first two adiabatic invariants could enhance particle losses.  However
one should mention that there are not observations of PC wave generation during HPS
impingements and this needs to be tested.  It is also uncertain how rapidly the relativistic electrons
would be lost by the above processes. It has been shown that the total loss of L >6.6 relativistic
electrons occurs in ~1 hr (Tsurutani et al., 2016b).


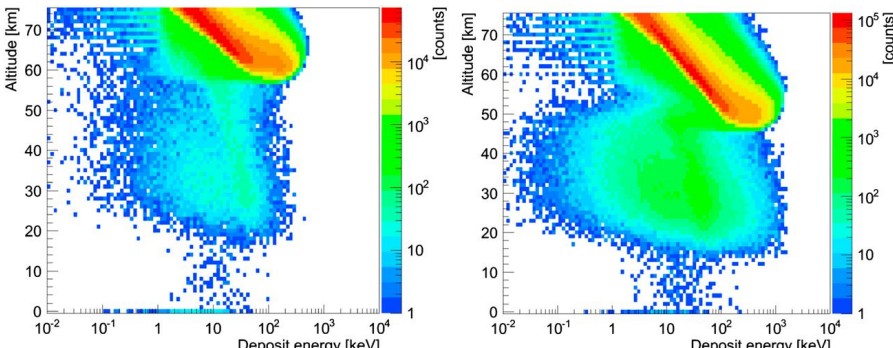


Figure 13. The GEANT4 code runs for the precipitation of E > 0.6 MeV electrons (left panel) and

E > 2.0 MeV electrons. The vertical scale is altitude above the ground and the horizontal scale is

energy deposition. The color scheme gives the amount of counts. Taken from Tsurutani et al.

(2016b).

Why can the loss of relativistic electrons to the atmosphere be important? Figure 13 shows the

results of the GEometry ANd Tracking 4 (GEANT4) code developed by the European

Organization for Nuclear Research (Agostinelli et al., 2003) applied to the relativistic electron

disappearance problem. The GEANT4 code takes into account Rayleigh scattering, Compton

scattering, photon absorption, gamma ray pair production, multiple scattering, ionization,

bremsstrahlung for electrons and positrons and annihilation of positrons (positron formation is not

germane for these "low energy" relativistic particles, but the code includes it anyway). A standard

atmosphere was used.

Figure 13 shows the GEANT4 Monte Carlo results for the electron shower for E > 0.6 MeV

electrons on the left and for E > 2.0 MeV electrons on the right. Two important features should be

noticed. First the bulk of energy deposition (the red areas) go down to ~60 km for the E > 0.6

MeV electron simulation and down to ~50 km for the E > 2.0 MeV electron simulation. This

portion of the energy from the incident electrons is due to direct ionization and particle energy

cascading. However there is a second region which might be extremely important. That is the

blue-green area that goes down to ~20 km for the E > 0.6 MeV simulation and ~16 km for the E >

2.0 MeV simulation. There are also "hits" seen on the ground. This lower altitude energy

deposition is due to the relativistic electrons interacting with atmospheric atomic and molecular

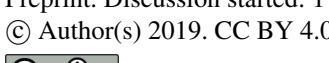



nuclei creating bremsstrahlung X-rays and γ-rays. X-rays and γ-rays have very large mean free
paths and thus can freely propagate through the dense atmosphere without interactions. They
propagate to much lower altitudes where they interact and have energy cascading again.

The reason why this process may be quite an important space weather topic is that it might relate
to atmospheric weather as well. Wilcox et al. (1973) discovered a correlation between
interplanetary HCS crossings and high atmospheric vorticity winds at 300 mb altitude. Over the
years a number of different explanations for the physics of the trigger has been offered (Tinsley
and Deen, 1991; Lam et al., 2013). Tsurutani et al. (2016) presented the above relativistic electron
dumping scenario (instead of HCS crossings) for the possible triggers of high atmospheric vorticity
winds. Quantitative estimates of potential energy deposition at different atmospheric altitudes
were provided in the original paper.

It is noted that the energy deposition should occur in a limited spatial region of the globe (just
inside the auroral zone and a small region of the dayside atmosphere) which is more geoeffective
than either cosmic ray energy or solar flare particle deposition. The fact that it is electron
precipitation gives an additional advantage that substantial energy is deposited at quite low
altitudes.

Advances to this problem can be made in a number of different ways. Simultaneous ground-
detected EMIC waves, γ-rays and atmospheric heating could be sought. Correlation with such
events with solar wind pressure pulses like the HPSs or interplanetary shocks (see Hajra and
Tsurutani, 2018b) would advance our knowledge of the details of such events.

Atmospheric heating events known as Sudden Stratospheric Warmings (SSWs) (Scherhag, 1960;
Harada et al., 2010) occur at subauroral latitudes by unknown causes. They are known to be related
to atmospheric wind system changes, perhaps the same phenomenon as the Wilcox et al. (1973)
effect. Atmospheric scientists generally assume that SSWs are created by gravity waves
propagating from lower atmosphere upward, but so far no one-to-one correlated case has been
found. Thus it would be quite interesting to see if space weather can have a major impact on




atmospheric weather. The connection between these two disciplines will be quite interesting for
the next generation of space weather scientists.

**3.2.6. Energetic particle precipitation and ozone depletion**

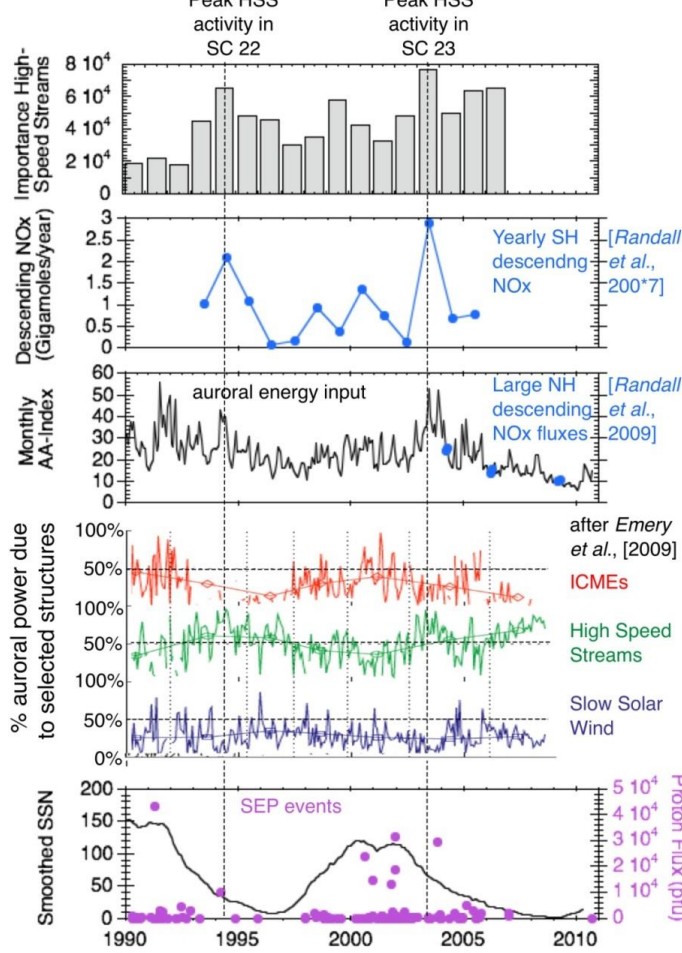


Figure 14.  The dashed vertical lines show the peaks in solar wind high speed streams during SC
22 and SC23.  These are coincident with the peaks in auroral energy input and the peaks in yearly
NOx descent.  We thank J.U. Kozyra for this unpublished figure.





Figure 14 shows two solar cycles of data, SC22 and SC23.  From top to bottom are the
"importance" of high speed streams, the descending NOx, the monthly AA index, the percent
auroral power due to three types of solar wind phenomena (ICMEs, HSSs and slow solar wind),
and the bottom panel solid line trace is the sunspot number (SSN). Also shown in the bottom panel
is the solar energetic particle (SEP) flux.

There are two vertical dashed lines.  They correspond to the peaks in HSS activity for SC22 and
SC23 (top panel), peaks in auroral energy input (third panel from the top), and peaks in the yearly
descending NOx (second panel from the top). It is noted that all three peaks are aligned in time.
The bottom panel shows that both dashed vertical lines correspond to times in the descending
phase of the solar cycle.

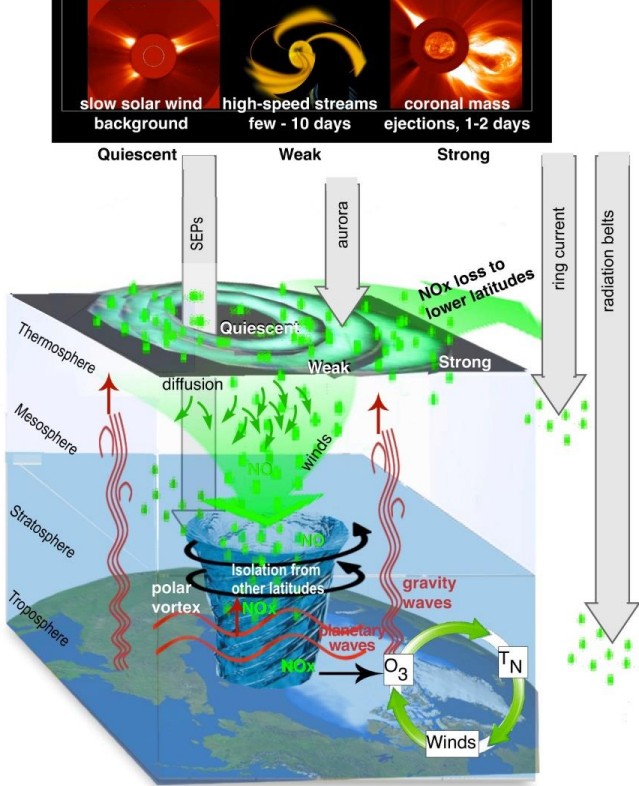






Figure 15.  The scenario for polar cap ozone destruction using the observations shown in Figure
27.  We thank J.U. Kozyra and her colleagues for this unpublished figure.

Figure 15 shows the Kozyra et al. (2015) scenario for ozone destruction over the polar cap.  The
top of the Figure shows the various types of solar wind that can affect atmospheric ozone. The
quiet solar wind will lead to quiescence.  HSSs lasting a few to ten days have weak effects and
ICMEs (and of course shock acceleration of energy particles) can have much strong effects.

The energetic particles associated with HILDCAAs and interplanetary shock acceleration will be
deposited in the near-polar regions of the both the north and south ionospheres.  Particles from
HILDCAA events will deposit their energy on closed auroral zone (~60° to 70°) magnetic field
lines. Solar energetic particles from interplanetary ICME shocks can propagate down the open
magnetic field lines of the polar caps. If the particles are energetic enough with sufficient gyroradii,
they can reach to as low latitudes as ~50° magnetic latitude.

The energetic particle entering the atmosphere lose a portion of their energy in the dissociation of
$N^2$ into N + N.  The nitrogen atoms will attach to oxygen atoms to form NOx. Auroral HILDCAA
~10 -100 kev energy particles will only penetrate to depths of ~75 km above the surface of the
Earth.   Solar energetic particles with greater kinetic energies can penetrate lower into the
atmosphere to ~50 to 60 km.  If there is a polar vortex, this vortex can "entrain" the NOx molecules
and atmospheric diffusion can bring them down to lower altitudes over months time duration. The
NOx can act as a catalyst in the destruction of ozone.

One interesting consequence of extreme ICME shocks is that one would expect extreme Mach
numbers to lead to both extreme SEP fluences and also extremely high energies. The former will
lead to greater production of NOx at the polar regions and the latter to deeper penetration and thus
less loss of NOx as they diffuse downward.  Alternatively there is a scenario where radiation belt
"killer" relativistic electrons can play an important role.  If there are large solar polar coronal holes
like in 1973-1975, HSSs could produce extremely intense and energetic relativistic electrons.
Shocks and HPS impingements on the magnetosphere could cause loss of the electrons to the lower
atmosphere.    This  magnetospheric  energy  pumping  and  dumping  may  have  important





consequences for NOx production. The topic of shock acceleration of energetic particles will be
discussed in more details in Section 4.1.

# 4.0. RESULTS: Interplanetary Shocks
**4.1. Interplanetary Shocks and Energetic Charged Particle Acceleration**

Interplanetary shocks have a variety of effects both in interplanetary space and the Earth's
magnetosphere. It is important for the reader to note that these space weather phenomena can occur
with or without the occurrence of magnetic storms. Shock and magnetic storm intensities are
related but only in a loose sense. The physical mechanism for energy transfer for different
pheomena is different. As one example, interplanetary shock acceleration of energetic charged
particles (called "solar cosmic rays") are due to an ICME ram energy driving the fast shocks which
then transfers energy to the charged particles. Solar cosmic ray events can occur with or without
magnetic storms (Halford et al. 2015, 2016; Mays et al., 2015; Foster et al. 2015). Some of the
major extreme space weather topics will be addressed below.

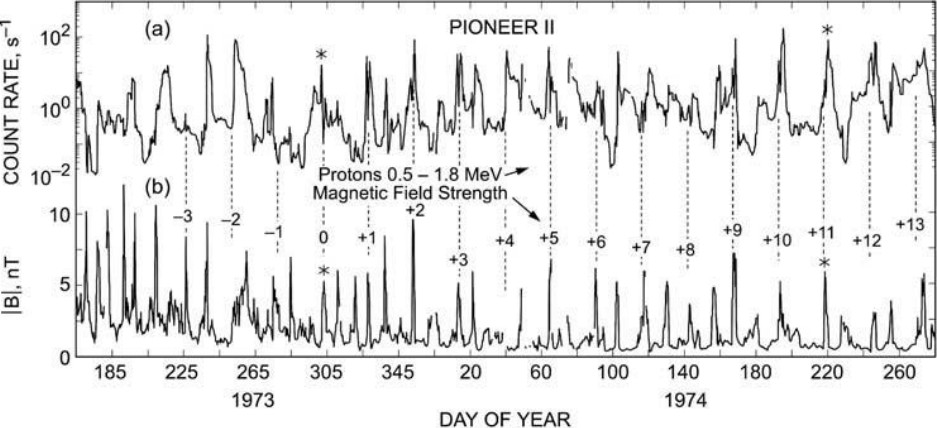


Figure 16. Energetic ~0.5 to 1.8 MeV protons accelerated by interplanetary fast forward and fast
reverse shocks. Taken from Tsurutani et al. (1982).

Acceleration of energetic particles in deep space was discovered by Pioneer 11 energetic particle
scientists (McDonald et al., 1976; Barnes and Simpson, 1976; Pesses et al., 1978, 1979; Van





Hollebeke et al., 1978; Christon and Simpson, 1979).  As the Pioneer 11 spacecraft traveled away
from the Sun, it was found that the particle fluences kept increasing, contrary to the concept of
adiabatic deceleration. The interplanetary magnetic field magnitude decreases with increasing
distance from the Sun, so one would expect energetic particle deceleration with distance. Thus it
was clear to scientists that something must be accelerating these particles in the interplanetary
medium.  Figure 16 shows one channel of the Pioneer 11 energetic proton count rate, ~0.5 to 1.8
MeV (see Simpson et al., 1974)  The bottom panel is the Pioneer 11 magnetic field (Smith et al.,
1975).  Some of the peak magnetic fields are numbered, corresponding to a ~25 day recurrence of
these magnetic structures.  The magnetic magnitude structures are identified as well-developed
CIRs (see Smith and Wolfe, 1976), bounded by fast forward and fast reverse shocks.

Tsurutani et al. (1982) identified the shocks and showed statistically that both forward and reverse
shocks were related to proton peak count rates.  One of the results, which still remains to be solved,
is that the proton peaks were generally higher at the reverse shocks.  What is the mechanism for
greater particle acceleration at fast reverse shocks? This has received little attention and should be
addressed in the future.

Reames (1999) has argued that fast forward shocks upstream (anti-solarward) of ICMEs are the
most important component of "solar flare" particle events.  Particle acceleration occurs throughout
interplanetary space from near the Sun (where the shocks first form) to 1 AU and beyond as the
shocks propagate through the heliosphere.  Studies of this acceleration as a function of longitudinal
distance away from magnetic connection to the flare site (this gives the variations in the shock
normal angle and thus dominant mechanism for acceleration—see Lee (2017) and references
therein) have been done by Lario (2012).  The features of the energetic particles in space have
different characteristics depending on these distances and the portion of the shock that the particles
are being accelerated from.

Forecasting the solar flare/interplanetary shock features such as the fluence, energy, spectra and
composition will require knowledge of the upstream seed population, upstream (and downstream)
waves, and shock properties such as the magnetosonic Mach number and shock normal angle.
This is a very difficult task since knowledge of the entire slow solar wind plasma from the Sun to

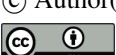



1 AU will be required for accurate forecasting. But again, the Parker Solar Probe and Solar Orbiter
may help in developing two points of measurements for modeling of specific events.

A more fundamental problem that can be answered by the Parker Solar Probe, Solar Orbiter and
ACE is why are interplanetary fast forward shock Mach numbers so low?  As previously
mentioned, Tsurutani and Lin (1985) from ISEE-3 measurements have found that at 1 AU, the
measured magnetosonic Mach numbers were typically only 1 to 3.  Tsurutani et al (2014) have
identified a shock with Mach number ~9 and Riley et al. (2016) has identified an event with
magnetosonic Mach number ~28.  The latter event was associated with the SOHO 2012 extreme
ICME which did not impact the Earth's magnetosphere.  The above are extreme events and little
or no events have been detected with intermediate values.

**4.2. Extreme Interplanetary Shocks and Extreme Interplanetary Energetic Particle**
**Acceleration**
Tsurutani and Lakhina (2014) have shown from simple calculations that with CME observed
speeds of 3,000 km/s (Yashiro et al., 2004; Gopalswamy, 2011), shock Mach numbers of ~45 are
possible. These Mach numbers are getting close to supernova shock numbers. Why haven't such
strong shocks been observed at 1 AU? If such events are possible, what would the energetic particle
fluences be? Experts on shock particle acceleration will hopefully answer this complex question.
It is well known that such solar flare particles enter the polar regions of the Earth's atmosphere
and cause radio blackouts. Will extreme solar flare particle fluence precipitation cause different
ionospheric effects other than those known today? This latter question might be addressed by
ionospheric modelers.

It should be noted that although space weather is a chain of events/phenomena going from the Sun
to interplanetary space to the magnetosphere, ionosphere and atmosphere, there is often not a direct
link between different facets of space weather.  Each feature of space weather should be examined
separately and it should not be assumed that an extreme flare will cause extreme cascading space
weather phenomena.  We use solar flare particles as an example for the readership.  The largest
solar flare particle event in the space age occurred in August 1972 (Dryer et al., 1976 and
references therein).  However there was no magnetic storm caused by the MC impact onto the





Earth's magnetosphere (the MC field direction was entirely northward, leading to geomagnetic
quiet: Tsurutani et al. 1992b). On the other hand, the largest magnetic storm on record is the
"Carrington" storm. The storm intensity will be discussed further in Section 7.0. There is little or
no evidence of large solar flare particle fluences in Greenland ice core data from that event (Wolff
et al., 2012; Schrijver et al., 2012). Usoskin and Kovfaltsov (2012) examining historical proxy
data ($^{14}$C and $^{10}$Be) also find a lack of any signature associated with the Carrington flare. Although
this is an extreme example, it is useful to mention it to illustrate the point.

An area that has received a lot of attention lately is ancient solar flares. Miyake et al. (2012)
discovered an anomalous 12% rapid increase in $^{14}$C content from 774 to 775 AD in Japanese cedar
tree rings. Usoskin et al. (2013) have argued that such an extreme radiation event could be
associated with an extreme solar energetic particle event (or a sequence of events). The latter
authors estimate the fluence of > 30 MeV particles was about 4.5 x $10^{10}$ cm$^{-2}$. Could such an
extreme particle event be associated with an extremely strong interplanetary shock or series of
shocks? Space weather scientists are currently working on this problem.

**4.3. Interplanetary shocks, dayside aurora and nightside substorms**

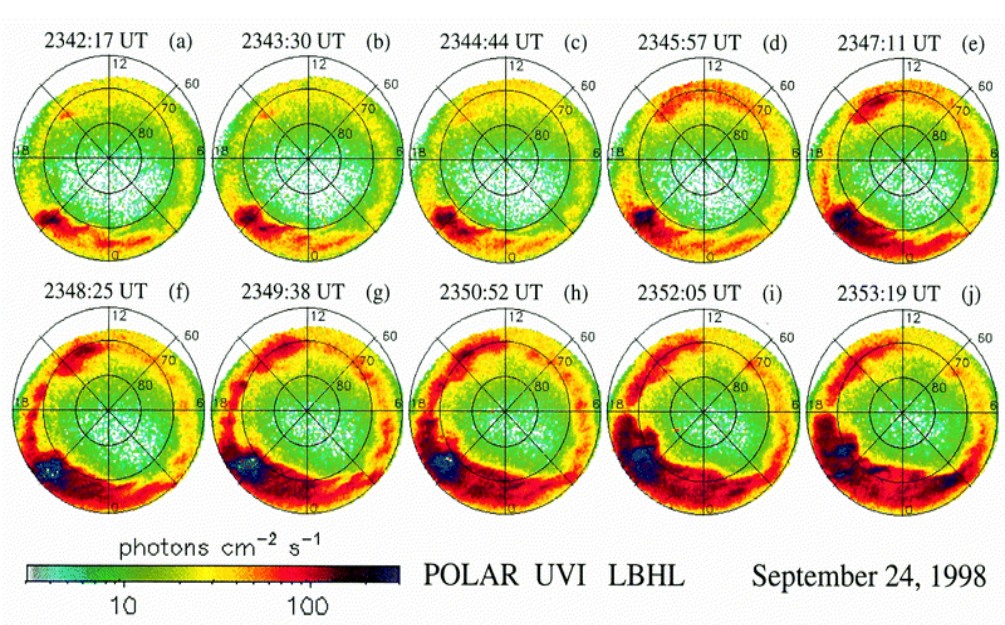






Figure 17. Interplanetary shocks cause dayside auroras and trigger nightside substorms. Northern
polar views of polar cap and auroral zones taken in UV wavelengths. Figure is taken from Zhou
and Tsurutani (2001).

Interplanetary shocks can trigger the precipitation of energetic ~10 to 100 keV electrons into the
auroral ionosphere (Halford et al. 2015). In fact, low energy (E < 10 keV) electron precipitation
can occur as well.  Figure 17 show interplanetary shock impingement auroral UV effects for an
event on September 23, 1998.  Each image has the north pole at the center and 60° magnetic
latitude (MLAT) shown at the outer edge.  Noon is at the top and dawn is at the right.  The cadence
between images is ~1min 13 s. From ACE measurements and propagation calculations it is known
that the fast forward shock arrived the magnetosphere between the images c), 2344:44 UT and d),
2345:47 UT. What is apparent in panel d) is the sudden appearance of aurora on the dayside (Zhou
and Tsurutani et al., 1999).  From further analyses of these shock auroral events, Zhou et al. (2003)
has shown that magnetospheric compression of preexisting ~10 to 100 keV electrons and protons
will generate both electromagnetic electron and proton plasma waves and diffuse auroras (as
discussed previously).  Also noted were the generation of field-aligned dayside currents.
Compression of the magnetosphere will generate Alfvén waves (Haerendel, 1994) which will
propagate along the magnetic fields lines down to the ionosphere.  Wave damping could provide
substantial ionospheric heating.

The mechanism for energy transfer from the solar wind to the magnetosphere is the absorption of
the solar wind ram energy.  Dayside auroras occur with shock impingement irrespective of the
interplanetary magnetic field Bz direction.  Another possible mechanism for the dayside aurora
not mentioned above are double layers above the ionosphere (Carlson et al., 1998) with the
acceleration of ~1 to10 keV electrons and the formation of discrete dayside auroras.  What is the
relative importance of these three different auroral energy mechanisms?  This would be an
excellent topic for the SWARM and Arase satellite missions.  Coordinated ground measurements
would be useful.

Returning back to Figure 17 panel e) 2347:11UT, there is a substorm intensification centered at
~2100 magnetic local time (MLT). The substorm further intensification and expansion can be



noted in the sequence of images. Interplanetary shock triggering of substorms has been known to
occur before the advent of imaging polar orbiting spacecraft (Heppner, 1955; Akasofu and Chao,
1980). The AE index had been used to identify these events.

Important topics are be where in the tail/magnetosphere does the substorm get initiated and by
what physical mechanism? Is it reconnection or plasma instabilities (Akasofu, 1972; Hones, 1979;
Lui et al., 1991; Lui, 1996; Baker et al., 1996; Lakhina, 2000)? Where does the energy come
from, recent solar wind inputs as suggested by Zhou and Tsurutani (1999) or stored tail energy or
even possibly solar wind ram energy (see Hajra and Tsurutani, 2018b)? The rapid response of the
magnetosphere to the shock should limit the downstream location of the substorm initiation point.
It should be noted that there are probably several different mechanisms for causing substorms.
Although this is only the shock triggering case, knowledge of this may help understand other cases.
The MMS mission will be ideally suited for addressing this question in the tail phase of the
mission.

**4.4. Interplanetary shocks and the formation of new radiation belts**



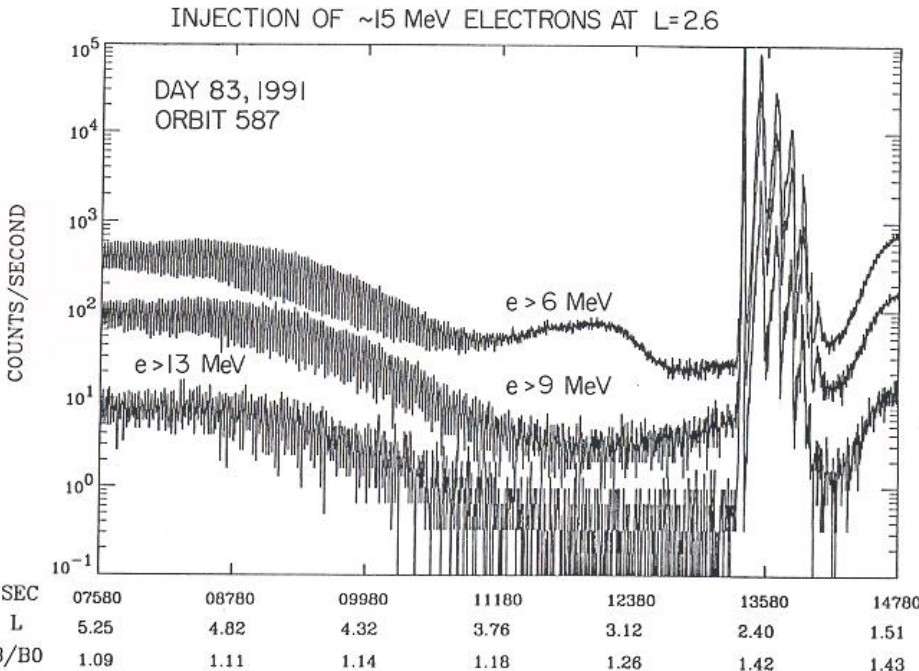

Figure 18. Shock creation of a new radiation belt in the magnetosphere. Figure taken from Blake et al. (1992).

Figure 18 shows evidence of a new "radiation belt" triggered by a strong interplanetary shock. The Figure shows three traces, E > 6 MeV, > 9 MeV and > 13 MeV fluences. At the time of the strong increase in all energy fluxes, the spacecraft was at L = 2.6. With increasing time, a second, then third, etc., pulse appear. These are "drift echos" where the energetic electrons gradient drift around the magnetosphere to return to the initial location.

**4.4.1. What is the mechanism to create this new radiation belt?**


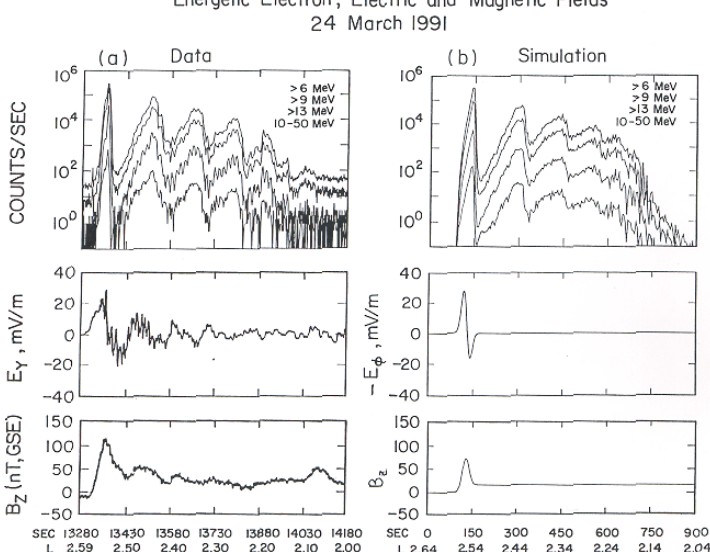

912

Figure 19. An expanded version of the relativistic electron pulse and measured magnetospheric
electric field and magnetic field Bz on the left and simulation results on the right. Taken from Li
et al. (2003).

916

The left hand column of Figure 19 show an expanded version of Figure 16 on the top with the
addition of the ~10 to 50 MeV count rate channel included. Next is the d.c. electric field in the Y
direction, and magnetospheric Bz on the bottom. The right hand column bottom shows a magnetic
pulse input into the system. This generates a time varying azimuthal electric field (right middle)
and the relativistic electron flux at the top right.

922

Using the input of a single magnetospheric magnetic pulse into the magnetosphere, Li et al. (2003)
simulated the acceleration and injection of E > 40 MeV electrons. What is interesting is that the
origin of the electrons was L > 6 with energies of only a few MeV. The reader should read Li et
al. (2003) for more details of the simulation and the results. Related works on acceleration of
magnetospheric electrons by shock impact on the magnetosphere can be found in Wygant et al.
(1994), Kellerman and Shprits, 2012; Kellerman et al., 2014; Foster et al. (2015).






How strong was the interplanetary shock? At this time it is because there were not any spacecraft
upstream of the Earth at the time of the event. However Araki (2014) has noted that this shock
caused a SI$^+$ of magnitude 202 nT. This is the second largest SI$^+$ in recorded history. In Tsurutani
and Lakhina (2014) with the assumption of a 3,000 km/s CME and only a 10% deceleration from
the Sun to 1 AU, they estimated a maximum SI$^+$ of 234 nT under normal conditions. Could this
1991 shock may have been close to the M =45 estimate mentioned earlier? However one cannot
really tell because the shock number strongly depends on the upstream plasma conditions which
can only be estimated. Also Tsurutani and Lakhina (2014) estimated a dB/dt six times larger than
the one used in the Li et al. (2003) modeling.   What would a maximum dB/dt cause in a new
radiation belt formation?

5.0. RESULTS: Solar Flares and Ionospheric Total Electron Content


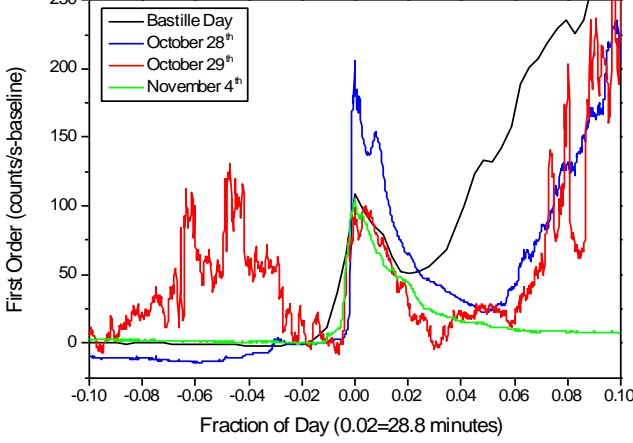


Figure 20.   The largest solar EUV flare in recorded history, October 28, 2003. Taken from
Tsurutani et al. (2005b).

Figure 18 shows four well- recognized solar X-ray flare events but taken in a narrow band 26-34
nm EUV spectrum.  This was done because some of the flare X-ray and EUV fluxes were so
intense that most spacecraft detectors became saturated (all except the SOHO SEM narrowband

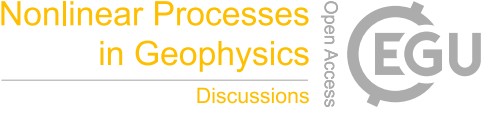



EUV detector). The flare intensities could only be estimated from fitting techniques for the
saturated data. Here we use the narrow band channel of the SOHO SEM detector where the four
above flares were not saturated. The four flare intensity profiles are the Bastille day (July 14,
2000) flare and three "Halloween" flares occurring on October 28, 29 and November 4, 2003. The
four flare count rate profiles were aligned so that they start at time zero. What is particularly
remarkable is that the October 28, 2003 flare has the highest EUV peak intensity of all four events
and was greater by a factor of ~2. This is the most intense EUV solar flare in recorded history.

After each flare reached a peak intensity and then decreased in count rate, there was often a
following increase in count rate. This is particularly notable in the Bastille day (black trace) flare.
This is contamination due to delayed energetic electrons reaching the spacecraft. The November
4 flare (green) did not have such contamination because it was a limb flare and presumably
(magnetic) connection from the flare site to the spacecraft was poor.

NOAA has estimated the November 4 flare as having an intensity of ~X28. This event saturated
the detector so this is a conservative estimate. Thomson et al. (2004) using a different technique
estimated a value of X45 for this event.

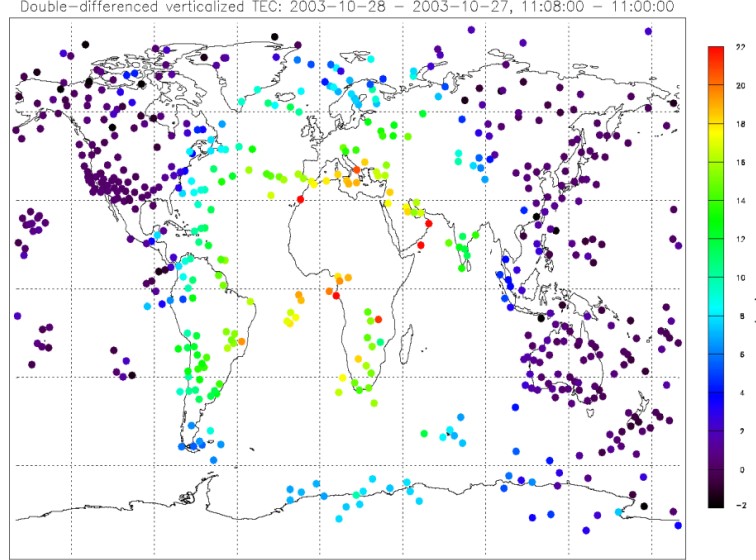




Figure 21.  The global TEC during the October 28, 2003 solar flare. The scale is given on the right.
The figure is taken from Tsurutani et al. (2005b).

Figure 21 shows the global total electron content (TEC) in the ionosphere after the October 28,
2003 solar flare.  The map has been adjusted so Africa, the subsolar point, is in the center of the
Figure.  The top and bottom of the plot correspond to the poles and the left side and right side
edges local midnight.  The enhanced TEC area corresponds to the sunlit hemisphere.  The nightside
hemisphere shows no TEC enhancement, as expected.  The TEC enhancement is due to ionization
by both X-rays, EUV photons and UV photons, all part of the solar flare spectrum.

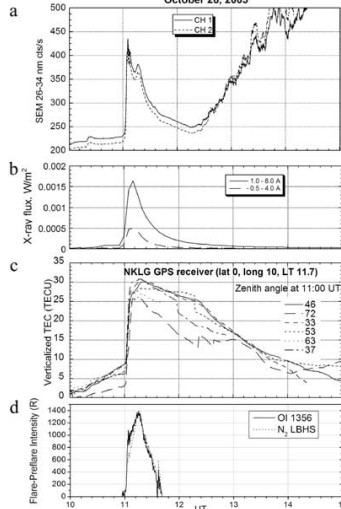


Figure 22.  The ionospheric and atmospheric effects of the October 28, 2003 solar flare.

Figure 22 shows the effects of the October 28 solar flare.  From top to bottom are the SOHO SEM
EUV count rate, the GOES X-ray flux, the Libreville, Gabon TEC data and the GUVI O and $N^2$
dayglow data. It is noted that the flare profiles in EUV and X-rays last ~tens of mins and are similar
in profile to each other.  However the TEC over Libreville last hours.  This is due to the EUV
portion of the solar flare.  These photons deposit their energy at ~170 to 220 km altitude where the
recombination time scales are ~ 3 to 4 hours.  Thus EUV photon ionization has longer lasting
ionospheric TEC effects.  The X-ray portion of the solar flare spectrum deposit their energy in the





~80 to 100 km altitude range where the recombination time scale is tens of min (Thomson et al.,
2005, and references therein).

Some future space weather problems are to understand if the solar flare spectrum varies and why
this happens. We have indicated that the 28 October 2003 and the 4 November 2003 flares were
significantly different. The question is why and how often does this happen? Ionospheric satellites
like the Constellation Observing System for Meteorology, Ionosphere and Climate-2 (COSMIC
II) and SWARM can probe for detailed altitude dependence of ionization to work backwards to
attempt to identify what spectrum would cause the layered ionization detected. Other questions are
how large can X-ray and EUV flares become? What will their ionospheric effects be?

## 6.0. RESULTS: Magnetic Storms and Prompt Penetrating Electric Fields
## (PPEFs)

For substorms, PPEFs occurring in the ionosphere have been known for a long time (Nishida and
Jacobs, 1962; Obayashi, 1967; Nishida, 1968; Kelley et al. 1979, 2003). In the last 10 years lots
of work was done on PPEFs during magnetic storms. Why didn't people look at storms earlier?
Because it was theoretically predicted that the PPEFs would be shielded out. Why doesn't
shielding happen? This is a very good question for future works.

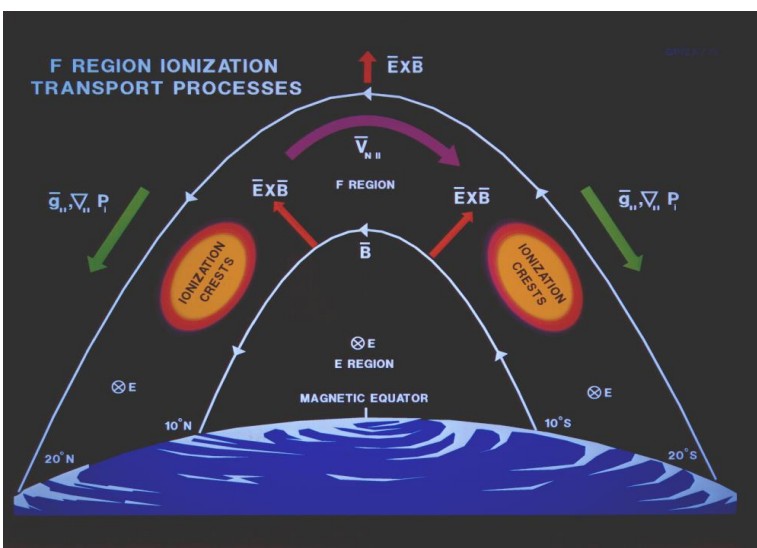

Figure 23. Dayside near equatorial ionization anomalies (EIAs) located ~± 10° on both sides of the magnetic equator. The local Earth magnetic field is shown in this schematic. The figure is taken from Anderson et al. (1996).

Figure 23 show the geometry of the Earth's magnetic field near the magnetic equator. It is parallel to the Earth's surface at the equator but where the equatorial ionization anomalies (EIAs) are located, the magnetic field is slanted. The EIAs are standardly located at ~±10° MLAT in the dayside magnetosphere. With red arrows, the figure also shows the direction of E x B convection. At exactly the magnetic equator, E x B is in a purely upward direction. At the positions of the EIAs, the E x B direction is both upward and to higher magnetic latitudes.

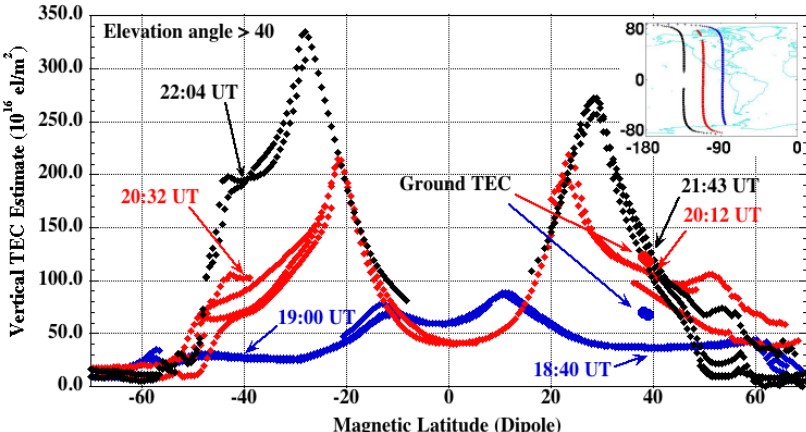


Figure 24. Three passes of the CHAMP satellite measuring the near equatorial and midlatitude
TEC during October 30, 2003.  CHAMP was at an altitude of ~430 km, so the TEC measured was
the thermal electron content above that altitude.  The figure is taken from Mannucci et al. (2005).

Figure 24 shows three passes of the CHAMP satellite in polar orbit with an altitude of ~430 km at
the near equatorial crossings.  The three orbits are given in the upper right hand portion of the
Figure.  The first TEC trace shown in blue is before the onset of the October 30-31 magnetic storm.
The two EIAs are identified by the TEC enhancements at ~ ±10° with peak intensities of ~80 TEC
units.   In the next pass (red trace), the EIAs are located at ~ ±21° MLAT and the peak intensities
are ~ 210 TEC units.  During the next satellite pass, the EIAs are located near ±30° and the TEC
values become as high as ~330 TEC units. This "movement" of the EIAs to higher magnetic
latitudes can be explained by a convective electric field (PPEF) in the east-west direction causing
an uplift to both EIAs by E x B convection as explained earlier associated with Figure 23.  However
why does the TEC increase to such high values?

The answer is as the PPEF removes the plasma from the ionospheric lower F region and brings it
to higher altitudes where recombination is long (hrs), the Sun's EUV photons replace the plasma
by photoionization of the upper atmosphere, replacing the lost plasma and thus increasing the "total
electron content" of the ionosphere. This is one cause of a "positive ionospheric storm".




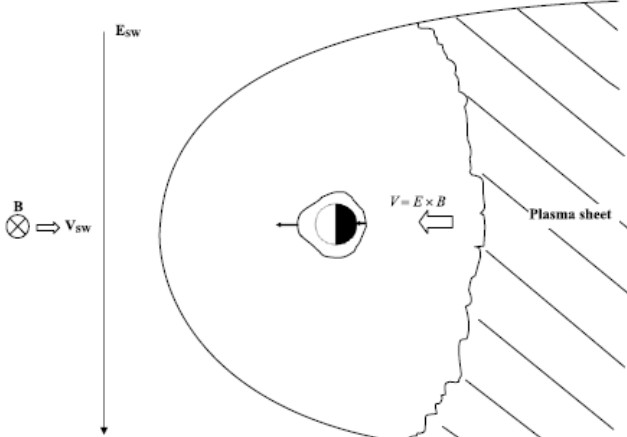


Figure 25. The interplanetary, magnetospheric and equatorial ionospheric electric fields during a
PPEF event. The Figure is taken from Tsurutani et al. (2004c; 2008b).

Figure 25 shows the interplanetary motional electric field for southward interplanetary Bz. The
electric field will be in the dawn-to-dusk direction. When magnetic reconnection takes place in
the nightside plasmasheet, the convective electric field will be in the same direction but with a
reduced amplitude. This electric field brings the plasmasheet plasma into the nightside low L
magnetosphere during magnetic storms. The PPEFs penetrate into the dayside equatorial
ionosphere (shown in Figure 24) and also the nightside equatorial ionosphere. However
significantly different from the dayside case, the E x B convection on the nightside will bring the
ionosphere to lower altitudes, leading to recombination and reduction in TEC. This is one form of
a "negative ionospheric storm". See Mannucci et al. (2005, 2008) for discussions of positive and
negative ionospheric storms.

There are many important questions about PPEFs which are almost always present during major
magnetic storms. As previously mentioned, "why aren't the electric fields shielded out?" What is
the mechanism for generating PPEFs, wave propagation from the polar ionosphere as suggested
by Kikuchi and Hashimoto (2016) or a more global picture as Figure 25 and Nishida and Jacobs
(1962) suggest? Figure 25 is a simple schematic. What are the real local time dependences of the
PPEF? Does this vary from storm to storm, and if so, why? Why does the PPEF magnitude vary




from one storm to the next? Again future spacecraft and ground based studies will be able to help
answer these questions.

## 7.0 RESULTS: The Carrington Storm

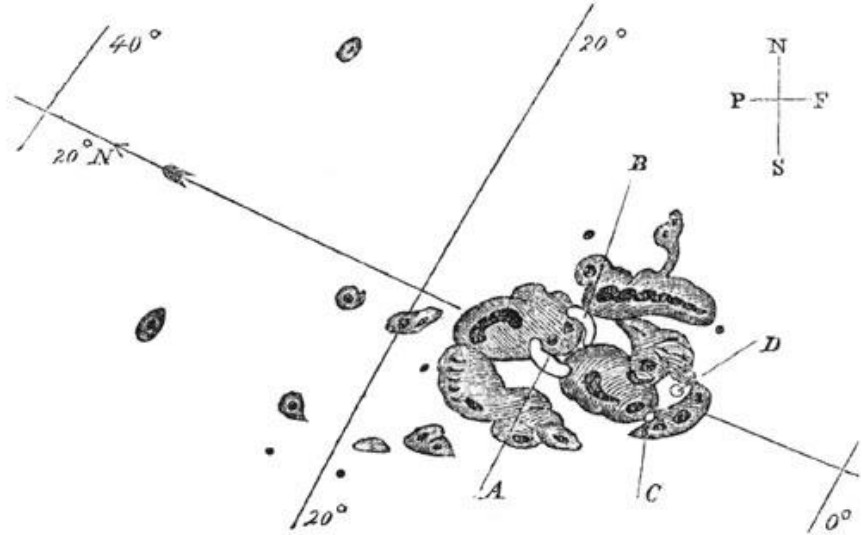


Figure 26. The solar active region during the Carrington 1 September 1859 optical solar flare. The
figure is taken from Carrington (1859).

Figure 26 is the active region (AR) that was hand-drawn by Richard Carrington.  This was the
source of the optical solar flare that he and Hodgson (1859) saw and reported on 1 September
1859.  See Cliver (2006) for a nice accounting of the observational activity taken during 1859 flare
interval and Kimball (1960) for an accounting of the aurora during the storm.  The optical part of
the flare lasted only ~ 5 min.  Some ~17 hr 40 min later a magnetic storm occurred at Earth
(Carrington, 1859).



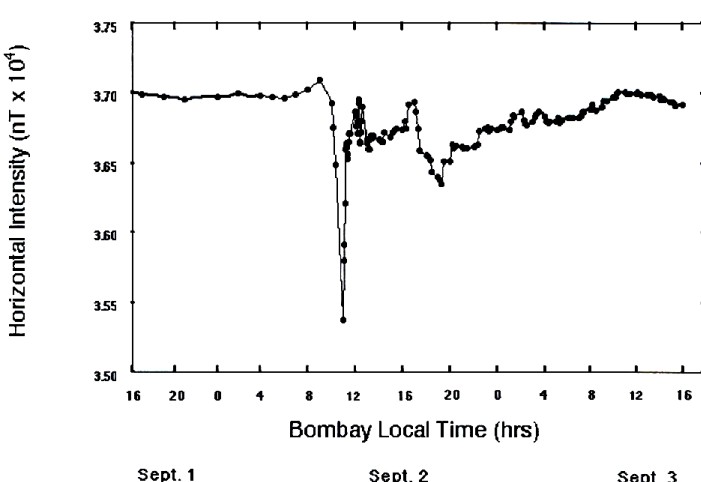


Figure 27. The Carrington storm detected in the Colaba, India magnetometer. The Figure is taken
from Tsurutani et al. 2003 and Lakhina et al. 2012.

Figure 27 shows the H-component magnetic field taken by the Colaba magnetic observatory during
the "Carrington" magnetic storm.  The $SI^+$ is estimated to be ~ 110 nT and the magnetic decrease
~1600 nT at Colaba (Mumbai, India).  The $SI^+$ and storm main phase has been recently shown to
be most likely caused by an upstream solar wind density of 5 particles $cm^{-3}$ and a MC with intensity
~90 nT (pointed totally southward) by Tsurutani et al. (2018a). No particularly unusual solar wind
conditions are believed to have been necessary, in contrast to the conclusions of Ngwira et al.

(2014).


The intensity of the "Carrington" storm was estimated as Dst =-1760 nT (Tsurutani et al., 2003)
based on observations of the lowest latitude of red auroras being at ±23° (Kimball, 1960). The
storm intensity was calculated using recent theoretical expressions of magnetospheric potentials
needed to convect plasma into such low latitudes.  Siscoe (1979) basing his estimate on a model
that treats the pressure as a constant along the magnetic flux tube came up with a value of Dst = -
2000 nT.





It should be mentioned that some researchers have taken exception with the Colaba magnetogram
as an indication of ring current effects (see Comment by Akasofu and Kamide (2005) and Reply
by Tsurutani et al. (2005a)). The Colaba magnetic profile is unlike those of ICME magnetic storms
discussed in Sections 2.3, 2.4 and 3.1 of this paper. Several researchers have estimated the storm
intensity based on the Colaba magnetogram (see articles in a special journal edited by Clauer and
Siscoe, 2006; Acero et al. 2018). The Colaba data clearly show that the storm had exceptionally
large geomagnetic effects, irregardless of the interpretation of the Colaba data.

The most accurate method of estimating a magnetic storm intensity is by using the latitude of the
aurora.  Red auroras (Stable Auroral Red or SAR arcs) are presumably an indication of the location
of the plasmapause (R.M. Thorne, private communication, 2002).  Kimball (1960) noted that "red
glows" were detected at $\pm23°$ from the geomagnetic equator during the Carrington event.  In 1960
the term "SAR arc" was not in use, but we can assume that this was what he was reporting. At the
present time, this is the most equatorward SAR arcs that have been observed (thus the most intense
magnetic storm).  That is until researchers find records of lower latitude auroras!

Comments on the short duration of the recovery phase has been made by Li et al. (2006).  A high
density filament was used to explain this unusual feature of the magnetic storm profile.  Tsurutani
et al. (2018a) have recently proposed another possibility.  During extreme events when the storm
time convection brings the plasmasheet into very low L, all of the standard ring current loss
processes will be enhanced.  There will be greater Coulomb scattering, greater charge exchange
losses and greater plasma wave growth with consequential greater wave-particle pitch angle
scattering and losses to the atmosphere.  Tsurutani et al. (2018a) focused particularly on wave-
particle interactions because the size of the loss cone will increase dramatically with decreasing L.
This, plus greater energetic particle compression due to inward convection, will lead to stronger
loss cone/temperature anisotropy instabilities, greater wave growth and thus greater losses.  This
hypothesis can be easily tested by magnetospheric spacecraft observations during large magnetic
storms and by magnetospheric modeling perhaps bringing some light to the unusual Colaba
magnetic signature.


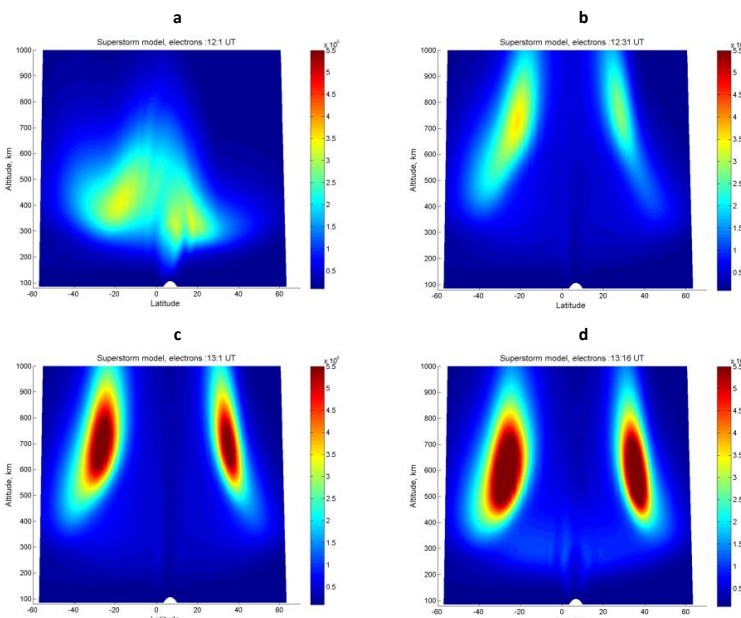


Figure 28. A model of the PPEF effects of the Carrington storm on the dayside ionosphere. The
input electric field was taken from Tsurutani et al. (2003) and the simulation was performed on
the Huba et al. (2000, 2002) SAMI2 code.  The figure is taken from Tsurutani et al. (2012).

**7.1. The Carrington PPEF**

One of the big worries for extreme space weather is the possible effects of PPEFs and the daytime
superfountain effect on the uplift of $O^+$ ions (positive ionospheric storms). Higher ion densities in
the exosphere will lead to the possibility of enhanced low altitude satellite drag.   In Tsurutani et
al. (2003), the authors used modern theories of the electric magnetospheric potential given by
Volland (1973), Stern (1975) and Nishida (1978) to determine the electric field during the
Carrington storm main phase.  The former authors obtained an estimate of ~20 mV/m.  They then
applied this electric field in the SAMI2 model with the results shown in Figure 26.

Figure 28 shows the SAMI2 results of the modeled dayside ionosphere with a ~20 mV/m added
to the diurnal variation electric field.  The quiet ionosphere is shown at the upper left.  The uplift
of the $O^+$ ions both in altitude and MLAT after ~30 min is given on the upper right panel.  The





maximum time that the electric field was applied was 1 hr.  The ionosphere at that time is shown
on the lower left.  The storm time equatorial ionospheric anomalies (EIAs) are located at |MLAT|
~30° to 40° and an altitude of ~550 to 900 km for the most dense portion of the EIAs. The bottom
right panel shows that the EIAs have come down in altitude but at higher latitudes ~15 min after
the termination of the PPEF application. Parts of the still intense EIAs are now beyond |MLAT| >
40° and now the bulk of the maximum density portion is at ~400 to 800 km altitude.

It was found that at altitudes of ~700 to 1,000 km, the $O^+$ densities are predicted to be 300 times
that of the quiet time neutral densities. It has been also been shown by Tsurutani and Lakhina
(2014) that in extreme cases, the magnetospheric/ionospheric electric field can be twice as large
as the Carrington storm and six times as large as the 1991 event.  Even if the magnetospheric
radiation belt is saturated (and there are other scientific papers that state that magnetospheric beta
can be greater than one: Chan et al. 1994; Saitoh et al. 2014; Nishiura et al., 2015), this is a different
facet of space weather and the electric field may not be saturated.   What will be the ionospheric
effects of these even larger electric fields?

A fundamental question for the future is "can the upward $O^+$ ion flow drag sufficient numbers of
oxygen neutrals upward so that the oxygen ions plus neutral densities are substantially higher?"
A short time interval analytic calculation done by Lakhina and Tsurutani (2017) and a mini-
Carrington event modeled by Deng et al. (2018) have indicated that the answer is "yes".  However
a full code needs to be developed and run to answer this question quantitatively.  This is an
interesting future problem for computer modelers.

**8.0 RESULTS:  Supersubstorms**


Superintense substorms (supersubstorms: SSSs) appear to be externally (solar wind) triggered.
Why are they important?  They might be the feature within extreme magnetic storms that cause
geomagneticall induced currents (GICs)/power outages.  This hypothesis needs to be tested.

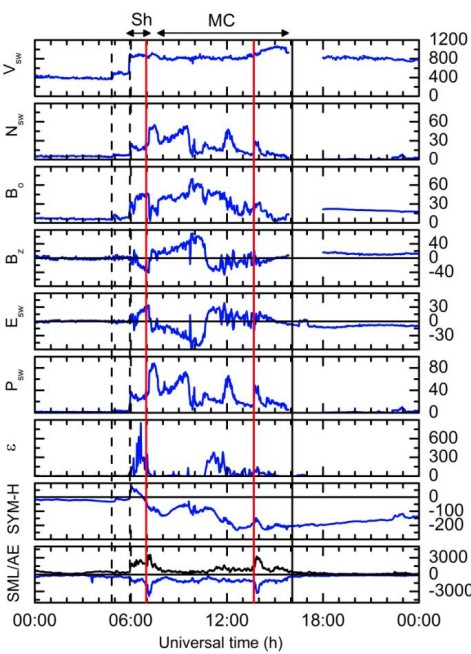


Figure 29. Two supersubstorms (SSSs) that occur during a two-phase magnetic storm. The onsets
of the supersubstorms are indicated by the vertical red lines. The figure is taken from Tsurutani et
al. (2015).

Figure 29 shows the solar wind data during an intense magnetic storm and two SSSs. From top to
bottom are the solar wind speed and density, the magnetic field magnitude and $B_z$ component, and
the interplanetary motional electric field, ram pressure and Akasofu epsilon parameter (Perreault
and Akasofu, 1978). The bottom two parameters are the SYM-H index and the SML index (blue)
and AE index (black). An initial forward shock is indicated by a vertical dashed line at ~0500 UT,
a second shock at ~0600 UT, and the two SSS onsets by red vertical lines. The criterion for a SSS
event was a SML peak value < -2500 nT (an arbitrary number, but chosen to be an extremely high
value). At the top of the diagram, the sheath region is indicated by a "Sh" and the magnetic cloud
region by "MC". The first storm main phase is caused by southward $B_z$ in the sheath and the
second, more intense main phase by southward $B_z$ in the MC.

It is noted that the SSS events in this case are not triggered at either of the two shocks nor do they
occur during the peak negative SYM-H values of the storm main phases. However the first SSS



event is collocated with a peak Esw and a peak southward Bz of the sheath plasma.  The SSS event
is also collocated with a large solar wind pressure pulse which is caused by an intense solar wind
density feature.  The second SSS event occurred in the recovery phase of the second magnetic
storm.  The IMF Bz was ~0 nT.   The second SSS event was associated with a solar wind pressure
pulse associated with a small density enhancement.

A study of SSSs from 1981 to 2012 was conducted by Hajra et al. (2016).  In that study a variety
of solar wind features were found to be associated with SSS onsets.  In that survey it was noted
that two SSS events were triggered by fast forward shocks.  One of these events will be discussed
below.

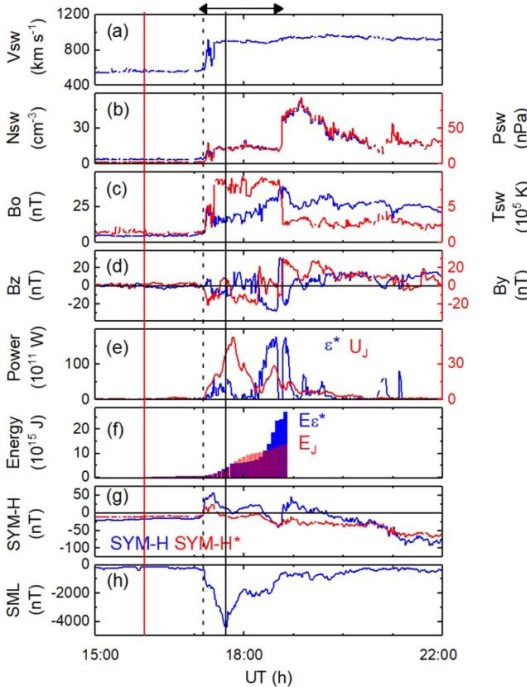


Figure 30. A SSS triggered by an interplanetary shock on 21 January 2005. The dashed vertical
line indicates a fast forward shock and the solid black line the peak intensity of the SSS event. The
figure is taken from Hajra and Tsurutani (2018b).





Figure 30 shows solar wind/interplanetary parameters and geomagnetic parameters during a SSS
event on 21 January 2005. From top to bottom are the solar wind speed, density and ram pressure,
the magnetic field magnitude and solar wind temperature (in the same panel), the IMF Bz and By
components (GSM coordinates), Joule energy and the Akasofu epsilon pressure corrected
parameter $\varepsilon^*$, the time-integrated energy input into the magnetosphere and time-integrated joule
energy. The next to the bottom panel contains the SYM-H* index and the pressure corrected SYM-
H index.  The bottom panel is the SML index. A dashed vertical line denotes the occurrence of a
fast forward shock.  A vertical solid line indicates the peak of the SSS event.

The SSS event onset at 1711 UT coincided with a shock with magnetosonic Mach number of ~5.5
with a shock normal angle of 81°. The high density sheath sunward of the shock causes a SI[+] of
~57 nT.  The solar feature associated with this event was an X7 class flare that occurred at ~0700
UT January 20 (Bombardieri et al., 2008; Saldanha et al., 2008; Pérez-Peraza et al., 2009; Wang
et al., 2009; Firoz et al., 2012; Bieber et al., 2013; Tan, 2013). The IMF Bz turned abruptly
southward at the time of the shock so this is part of the energy driving the event. When the IMF
Bz turned abruptly northward at ~1738 UT, the SSS began a recovery phase.  This was followed
by a solar filament (Kozyra et al., 2013) but the latter was not geoeffective in this case.


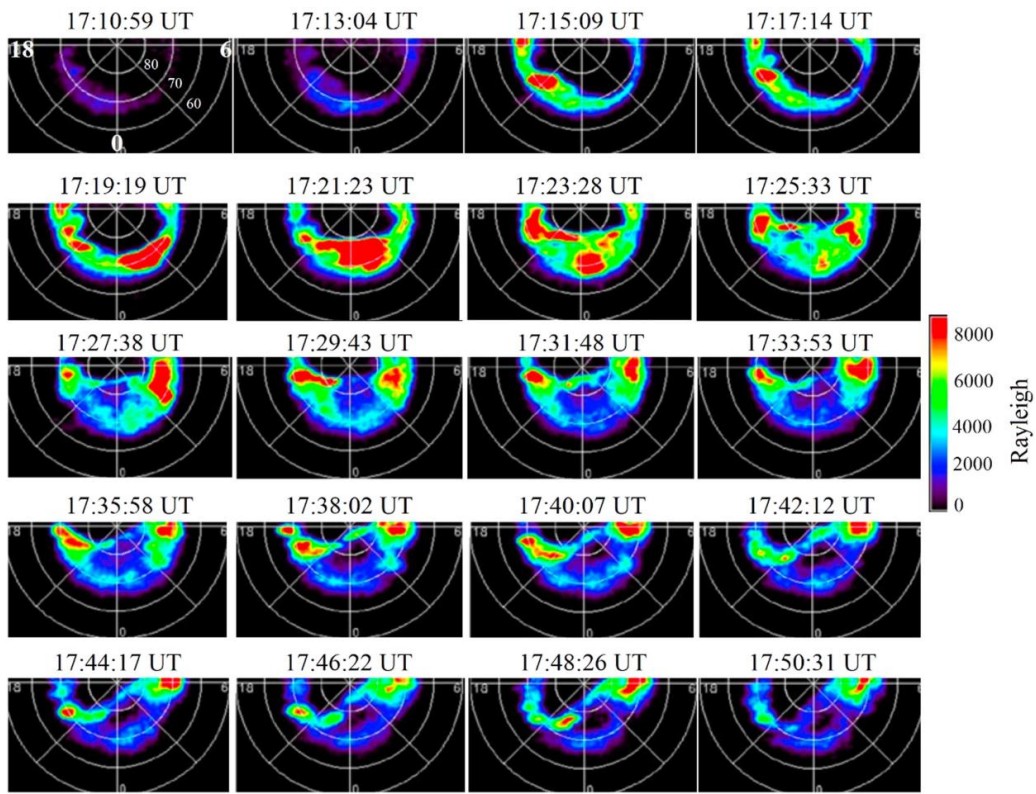

Figure 31. IMAGE-FUV images taken from ~1711 UT to ~1751 UT on January 21. These selected auroral images correspond to the SSS event in Figure 30.

Figure 31 contains the Imager for Magnetopause-to-Aurora global Exploration (IMAGE) far ultraviolet images for the SSS event in Figure 30. At ~1713 UT there was a small brightening at ~68° MLAT, which was a very small substorm or pseudobreakup (Elvey, 1957; Tsurutani et al., 1998; Aikio et al., 1999). At ~1715 UT, 2 min later there was a ~2100 MLT premidnight brightening of the aurora at ~68° to 75°. At ~1719 UT the most intense aurora was located at ~68° to 72° in the postmidnight/morning sector, ~0000 to 0400 MLT. The aurora moved from a dominant premidnight location to a postmidnight location in ~4 min.

By ~1726 UT there was almost no aurora of significant intensity at local midnight. At the peak of the SML value at ~1738 UT until ~1751, there were both intense premidnight and postmidnight auroras.






The SSS event did not exhibit the Akasofu (1964) standard model of a substorm with an
intensification at midnight and then expansion to the west, east and north. The changes in the
location of intense auroras were too rapid to track with the IMAGE cadence of ~2 min.

The SSS events display rapid auroral movements which may entail the appearance of sudden local
field-aligned currents. Even smooth motion of auroral forms will cause strong dB/dt effects over
local ground stations. SSS events may be features that can cause GIC effects that have been
attributed to "magnetic storms". Thus it might be the SSS events within magnetic storms which
are the real cause. SWARM satellites are excellently instrumented spacecraft that can study the
SSS events in detail and possible resultant GIC effects. However as noted in the auroral images,
there is a need for even higher time resolution global images than is present today.

## 8.0. CONCLUSIONS: Forecasting Space Weather

We have discussed the current knowledge about various facets of space weather. There are others
which we have not touched upon because of limited time and knowledge. The reader should know
that other areas of space weather exist which may be equally important. Perhaps other reviews
will cover those topics.

The most critical area for forecasting magnetic storms, either during solar maximum or the
declining phase of the solar cycle is the prediction of the magnetic field Bz in front of the Earth's
magnetosphere. For CME storms (primarily during solar maximum), this is identifying MC Bz
near the Sun and understanding and predicting the evolution of the MC from the Sun to Earth.
This major challenge will be applicable for the prediction of extreme magnetic storms and
hopefully great progress will be made in the next 5 to 10 years.

For sheaths upstream of ICMEs during solar maximum and CIRs during the declining phase, the
problem is different. Detailed knowledge of the slow solar wind in the space between the Sun and
Earth are needed to accurately describe and predict the IMF Bz that impacts the Earth. So far little

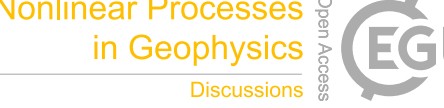



work has been applied towards predicting the slow solar wind (plus verification). Effort needs to
be placed in this area to be able to forecast intense to moderate magnetic storms.

A great deal of knowledge presently exists for establishing SEP events, those energetic particles
associated with acceleration at ICME shock fronts.  What is needed for better forecasting is to
understand the Mach number of the shocks, the shock normal angles and possibly upstream "seed"
particles.  The upstream seed particle population is similar to the sheath Bz problem in that this
component of the slow solar wind needs to be modeled more carefully.  Three spacecraft in the
solar wind at different distances from the Sun should help a lot.

The appearance of HSSs at 1 AU is a very tractable problem.  That is if the coronal hole boundaries
in the photosphere can be established firmly and the HSS propagation to 1 AU can be done
accurately.  However the most difficult task again is the IMF Bz.  If Alfvén waves are generated
in the interplanetary medium, this will make the task even more difficult.  One solution is to
measure the interplanetary magnetic field at 1 AU and use filtering techniques (Guarnieri et al.
2018) or again have large apogee Earth orbiters like the IMP-8 spacecraft again.  Another
possibility is developing some type of statistical IMF Bz generator.

Predicting the interplanetary shock Mach numbers and ram pressure jumps will allow
foreknowledge of new radiation belt formation, SI[+] effects and magnetospheric and ionospheric
dB/dt effects.  Dayside auroral intensities and nightside substorm triggering will also be enhanced
by predicting oncoming shocks.

Several spacecraft missions have been mentioned in relationship to some forecasting problems.
However the reader should note that the missions and/or their data alone will not solve these
problems.  It will be the scientists either on these missions or perhaps totally independent scientists
who will make the most progress on these problems. An example is magnetic storms caused by
interplanetary shocks/sheaths and CIRs.  How long will it take scientists to be able to accurately
forecast the time of occurrence of the storm (the easiest part) and the intensity (the hardest part)?
Here we will not make an estimate of how long this will take.  Shock acceleration of solar flare
particles is clearly a fundamental part of space weather.  How long will it scientists to be able to





predict the fluence and spectral shape at a variety of distances away from the Sun? This is a
fundamental problem which space agencies are not currently directly addressing.



# 10.0 GLOSSARY
**Partially taken from:** *"From the Sun: Auroras, Magnetic Storms, Solar Flares, Cosmic*
*Rays"* (Suess and Tsurutani, 1998, AGU Press)

**Adiabatic Invariant**: In a nearly collisionless, ionized gas, electrically charged particles orbit
around magnetic lines of force. Certain physical quantities are approximately constant for
slow (adiabatic) changes of the magnetic field in time or in space and these quantities are
called *adiabatic invariants*. For example, the magnetic moment of a charged particle,
$\mu = mV\perp^2/(2B)$, is such a constant where $V\perp$ is the velocity of the particle perpendicular to
the magnetic field, B is the magnetic field strength, and m is the particle mass. In a
converging field such as in approaching the pole of a dipole magnetic, the field strength
increases and therefore $V\perp$ increases as well because $\mu$ has to remain constant.
**Aeronomy**: The science of the (upper) regions of atmospheres, those regions where dissociation
of molecules and ionization are present.
**Alfvén Wave** (magnetohydrodynamic shear wave): A transverse wave in magnetized plasma
characterized by a change of direction of the magnetic field with no change in either the
intensity of the field or the plasma density.
**Anisotropic Plasma**: A Plasma whose properties vary with direction relative to the ambient
magnetic field direction. This can be due, for example, to the presence of a magnetic or
electric field. See also Isotropic Plasma; Plasma.
**Arase satellite, formerly called Exploration of energization and Radiation in Geospace** or
**ERG**: a scientific satellite developed by the Institute of Space and Astonautical Science





(ISAS) of the Japanese Aerospace Exploration Agency (JAXA) to study the Van Allen
radiation belts.
**Astronomical Unit (AU)**: The mean radius of the Earth's orbit, 1.496 x10$^{13}$ cm.
**Aurora**: A visual phenomenon that occurs mainly in the high-latitude night sky. Auroras occur
within a band of latitudes known as the auroral oval, the location of which is dependent
on the intensity of geomagnetic activity. Auroras are a result of collisions between
atmospheric gases and charged particles (mostly electrons) precipitating from the outer
parts of the magnetosphere and guided by the geomagnetic field. Each gas (oxygen and
nitrogen molecules and atoms) emits its own when bombarded by the precipitating
particles. Since the atmospheric composition varies with altitude, and the faster
precipitating particles penetrate deeper into the atmosphere, certain auroral colors
originate preferentially from certain heights in the sky. The auroral altitude range is 80 to
500 km, but typical auroras occur 90 to 250 km above the ground. The color of the
typical aurora is yellow-green, from a specific transition line of atomic oxygen. Auroral
light from lower levels in the atmosphere is dominated by blue and red bands from
molecular nitrogen and molecular oxygen. Above 250 km, auroral light is characterized
by a red spectral line of atomic oxygen. To an observer on the ground, the combined light
of these three fluctuating, primary colors produces an extraordinary visual display.
Auroras in the Northern Hemisphere are called the aurora borealis or "northern lights".
Auroras in the Southern Hemisphere are called aurora australis. The patterns and forms of
the aurora include quiescent "arcs", rapidly moving "rays" and "curtains," "patches," and
"veils."
**Auroral Electrojet (AE)**: See Electrojet.
**Auroral Oval**: An elliptical band around each geomagnetic pole ranging from about 75 degrees
magnetic latitude at local noon to about 67 degrees magnetic latitude at midnight under
average conditions. It is the locus of those locations of the maximum occurrence of
auroras, and widens to both higher and lower latitudes during the expansion phase of a
magnetic substorm.





1358 **Beta** (e.g., low-beta plasma)**:** The ratio of the thermal pressure to the magnetic 'pressure' in a

1359   plasma - p/ $(B^2/(8\pi))$ in centimeter-gram-second (c.g.s.)

1360 **Bow Shock** (Earth, heliosphere): A collisionless shock wave in front of the magnetosphere

1361   arising from the interaction of the supersonic solar wind with the Earth's magnetic field.

1362   An analogous shock is the heliospheric bow shock which exists in front of the

1363   heliosphere and is due to the interaction of the interstellar wind with the solar wind and

1364   the inter planetary magnetic field.

1365 **Charge Exchange**: An interaction between a charged particle and a neutral atom wherein the

1366   charged particle becomes neutral and the neutral particle becomes charged through the

1367   exchange of an electron.

1368 **Cloud** (magnetic): see Magnetic Cloud.

1369 **Collisional** (de-) **Excitation**: Excitation of an atom or molecule to a higher energy state due to a

1370   collision with another atom, molecule, or ion. The higher energy state generally refers to

1371   electrons in higher energy around atoms.  Deexcitation is the reduction of a higher

1372   electron energy state to a lower one, usually accomplished by a collision with another

1373   atom, molecule or ion.

1374 **Convection** (magnetospheric, plasma, thermal): The bulk transport of plasma (or gas) from one

1375   place to another, in response to mechanical forces (for example, viscous interaction with

1376   the solar wind) or electromagnetic forces. Thermal convection, due to heating from below

1377   and the gravitational field, is what drives convection inside the Sun.  Magnetospheric

1378   convection is driven by the dragging of the Earth's magnetic field and plasma together by

1379   the solar wind when the magnetic field becomes attached to the magnetic field in the

1380   solar wind.

1381 **Coriolis Force**: In the frame of a rotating body (such as the Earth), a force due to the bodily

1382   rotation. All bodies that are not acted upon by some force have the tendency to remain in

1383   a state of rest or of uniform rectilinear motion (Newton's First Law) so that this force is

1384   called a "fictitious" forces. It is a consequence of the continuous acceleration which must

1385   be applied to keep a body at rest in a rotating frame of reference.





**Corona**: The outermost layer of the solar atmosphere, characterized by low densities ($<10^9$ cm$^{-3}$
or $10^{15}$ m$^{-3}$) and high temperatures ($>10^6$ K).

**Coronal Hole**: An extended region of the solar corona characterized by exceptionally low
density and in a unipolar photospheric magnetic field having "open" magnetic field
topology. Coronal holes are largest and most stable at or near the solar poles, and are a
source of high speed (700-800 km/s) solar wind. Coronal holes are visible in several
wavelengths, most notably solar x-rays visible only from space, but also in the He 1083
1393          nm line which is detectable from the surface of the Earth. In soft x-ray images (photon
energy of ~0.1-1.0 keV or a wavelength of 10-100 Å), these regions are dark, thus the
name "holes".

**Coronal Mass Ejection (CME):** A transient outflow of plasma from or through the solar
corona. CMEs are often but not always associated with erupting prominences,
disappearing solar filaments, and flares.

**Corotation** (with the Earth): A plasma in the magnetosphere of the Earth is said to be corotating
with the Earth if the magnetic field drags the plasma with it and together they have a 24
1401          hour rotation period.

**Cosmic Ray** (galactic, solar): Extremely energetic (relativistic) charged particles or
electromagnetic radiation, primarily originating outside of the Earth's magnetosphere.
Cosmic rays usually interact with the atoms and molecules of the atmosphere before
reaching the surface of the Earth. The nuclear interactions lead to formation of daughter
products, and they in turn to granddaughter products, etc.,; thus there is a chain of
reactions and a "cosmic ray shower" Some cosmic rays come from outside the solar
system while others are emitted from the Sun in solar flares. See also Anomalous Cosmic
Ray; Energetic Particle; Solar Energetic Particle (SEP) Event.

**Constellation Observing System for Meteorology, Ionosphere and Climate-2 (COSMIC II):**
A joint Taiwan National Space Organization (NSPO)-U.S. National Oceanic and Atmospheric
Administration (NOAA) mission of six satellites in low-inclination orbit to study the Earth's
ionosphere.





**Corotating Interaction Region (CIR):** An interplanetary region of high magnetic fields and
plasma densities created by the interaction of a high speed solar wind stream with the upstream
slow solar wind. The antisunward portion of the CIR is compressed slow solar wind plasma and
magnetic fields, and the sunward portion is compressed fast solar wind plasma and magnetic
fields. The two regions of the CIR are separated by a tangential discontinuity.

**Cyclotron Frequency**: When a particle of charge q moves in a magnetic field B, the particle
orbits, or gyrates around the magnetic field lines. The cyclotron frequency is the
frequency of this gyration, and is given by $\omega_c = q|\mathbf{B}|/mc$, where m is the mass of the
particle, and c is the velocity of light (in centimeter-gram-second (c.g.s.) units).
**Cyclotron Resonance**: The frequency at which a charged particle experiences a Doppler-shifted
wave at the particle's cyclotron frequency. Because the particle and wave may be
traveling at different speeds and in different directions, there is usually a Doppler shift
involved.
**D Region**: A daytime region of the Earth's ionosphere beginning at approximately 40 km,
extending to 90 km altitude. Radio wave absorption in this region can be significantly
increased due to increasing ionization associated with the precipitation of solar energetic
particles through the magnetosphere and into the ionosphere.
**Diffusion**: The slow, stochastic motion of particles.
**Diffusive Shock Acceleration**: Charged particle acceleration at a collisionless shock due to
stochastic scattering processes caused by waves and plasma turbulence. See also Shock
Wave (collisionless).
**Dipole Magnetic Field**: A magnetic field whose intensity decreases as the cube of the distance
from the source. A bar magnet's field and the magnetic field originating in the Earth's
core are both approximately dipole magnetic fields.
**Drift** (of ions/electrons): As particles gyrate around magnetic field lines, their orbits may "drift"
perpendicular to the local direction of the magnetic field. This occurs if there is a force





also perpendicular to the field - e.g. an electric field, curvature in the magnetic field
direction, or gravity.
**Driver Gas**: A mass of plasma and entrained magnetic field that is ejected from the Sun, that has
a velocity higher than the upstream plasma, and which "drives" a (usually collisionless)
shock wave ahead of itself. The magnetic cloud within an ICME is the same thing as a
driver gas.
**Dst Index**: A measure of variation in the geomagnetic field due to the equatorial ring current. It
is computed from the H-components at approximately four near-equatorial stations at
hourly intervals. At a given time, the Dst index is the average of variation over all
longitudes; the reference level is set so that Dst is statistically zero on internationally
designated quiet days. An index of -50 nT (nanoTesla) or less indicates a storm-level
disturbance, and an index of -200 nT or less is associated with middle- latitude auroras.
Dst is determined by the World Data Center C2 for Geomagnetism, Kyoto University,
Kyoto, Japan.
**Dynamo** (solar magnetospheric): The conversion of mechanical energy (rotation in the case of
the Sun) into electrical current. This is the process by which magnetic fields are amplified
by the induction of plasmas being forced to move perpendicular to the magnetic field
lines. See also Mean Field Electro-Dynamics.
**E-Region**: A daytime region of the Earth's ionosphere roughly between the altitudes of 90 and
160 km. The E-region characteristics (electron density, height, etc.) depend on the solar
zenith angle and the solar activity. The ionization in the E layer is caused mainly by x-
rays in the range 0.8 to 10.4 nm coming from the Sun.
**Ecliptic Plane**: The plane of the Earth's orbit about the Sun. It is also the Sun's apparent annual
path, or orbit, across the celestial sphere.
**Electrically Charged Particle:** Electrons and protons, for example, or any atom from which
electrons have been removed to make it into a positively charged ion. The elemental
charge of particles is $4.8 \times 10^{-10}$ esu. An electron and proton have this charge. Combined (a
hydrogen atom), the charge is zero. Ions have multiples of this charge, depending on the
number of electrons which have been removed (or added).





**Electrojet**: (1) Auroral Electrojet (AE): A current that flows in the ionosphere at a height of ~100 km in the auroral zone. (2) Equatorial Electrojet: A thin electric current layer in the ionosphere over the dip equator at about 100 to 115 km altitude.

**Electron Plasma Frequency/Wave**: The natural frequency of oscillation of electrons in a neutral plasma (e.g., equal numbers of electrons and protons).

**Electron Volt (eV)**: The kinetic energy gained by an electron or proton being accelerated in a potential drop of one Volt.

**ESA**: European Space Agency

**Extreme Ultraviolet (EUV)**: A portion of the electromagnetic spectrum from approximately 10 to 100 nm.

**Extremely Low Frequency (ELF)**: That portion of the radio frequency spectrum from 30 to 3000 Hz.

**Fast Mode** (wave/speed): In magnetohydrodynamics, the fastest wave speed possible. Numerically, this is equal to the square root of the sum of the squares of the Alfvén speed and plasma sound speed.

**Field Aligned Current**: A current flowing along (or opposite to) the magnetic field direction.

**Filament**: A mass of gas suspended over the chromosphere by magnetic fields and seen as dark ribbons threaded over the solar disk. A filament on the limb of the Sun seen in emission against the dark sky is called a prominence. Filaments occur directly over magnetic-polarity inversion lines, unless they are active.

**Flare**: A sudden eruption of energy in the solar atmosphere lasting minutes to hours, from which radiation and energetic charged particles are emitted. Flares are classified on the basis of area at the time of maximum brightness in H alpha.

Importance 0 (Subflare): < 2.0 hemispheric square degrees

Importance 1: 2.1-5.1 square degrees

Importance 2: 5.2-12.4 square degrees





1497       Importance 3: 12.5-24.7 square degrees

1498       Importance 4: >= 24.8 square degrees

1499       [One square degree is equal to $(1.214 \times 10^4$ km) squared = 48.5 millionths of the

1500       visible solar hemisphere.] A brightness qualifier F, N, or B is generally appended

1501       to the importance character to indicate faint, normal, or brilliant (for example,

1502       2B).

1503   **Flux Rope**: A magnetic phenomenon which has a force-free field configuration.

1504   **Force Free Field**: A magnetic field which exerts no force on the surrounding plasma. This can

1505       either be a field with no flowing electrical currents or a field in which the electrical

1506       currents all flow parallel to the field.

1507   **Free Energy** (of a plasma): When an electron or ion distribution is either non-Maxwellian or

1508       anisotropic, they are said to have free energy" from which plasma waves can be

1509       generated via instabilities. The waves scatter the particles so they become more isotropic,

1510       reducing the free energy.

1511   **Frozen-in Field**: In a tenuous, collisionless plasma, the weak magnetic fields embedded in the

1512       plasma are convected with the plasma. i.e., they are "frozen in."

1513   **Galactic Cosmic Ray (GCR)**: See Cosmic Ray.

1514   **Gamma Ray**: Electromagnetic radiation at frequencies higher than x-rays.

1515   **Geomagnetic Storm**: A worldwide disturbance of the Earth's magnetic field, distinct from

1516       regular diurnal variations. A storm is precisely defined as occurring when $D_{sT}$ becomes

1517       less than -50 nT (See geomagnetic activity).

1518       Main Phase: Of a geomagnetic storm, that period when the horizontal magnetic field at

1519       middle latitudes decreases, owing to the effects of an increasing magnetospheric ring

1520       current. The main phase can last for hours, but typically lasts less than 1 day.

1521       Recovery Phase: Of a geomagnetic storm, that period when the depressed northward field

1522       component returns to normal levels. Recovery is typically complete in one to two days.





**Geosynchronous Orbit**: Term applied to any equatorial satellite with an orbital velocity equal to

the rotational velocity of the Earth. The geosynchronous altitude is near 6.6 Earth radii

(approximately 36,000 km above the Earth's surface). To be geostationary as well, the

satellite must satisfy the additional restriction that its orbital inclination be exactly zero

degrees. The net effect is that a geostationary satellite is virtually motionless with respect

to an observer on the ground.

**GeV**: $10^9$ electron Volts (Giga-electron Volt).
**Global Positioning System** (GPS): is a global navigation satellite system that provides

geolocation and time information to a GPS receiver anywhere on or near the Earth where

there is an unobstructed line of sight to four or more GPS satellites.

**Global-scale Observations of the Limb and Disk (GOLD):** a NASA mission to " investigate

the dynamic intermingling of space and Earth's uppermost atmosphere"

**Heliosphere**: The magnetic cavity surrounding the Sun, carved out of the galaxy by the solar

wind.

**Heliospheric Current Sheet (HCS)**: This is the surface dividing the northern and southern

magnetic field hemispheres in the solar wind. The magnetic field is generally one polarity

in the north and the opposite in the south so just one surface divides the two polarities.

However, the Sun's magnetic field changes over the 11-year solar sunspot cycle and

reverses polarity at solar maximum. The same thing happens in the magnetic field carried

away from the Sun by the solar wind so the HCS only lies in the equator near solar

minimum. It is called a "current sheet" because it carries an electrical current to balance

the oppositely directed field on either side of the surface. It is very thin on the scale of the

solar system - usually only a few proton gyroradii, or less than 100,000 km.

**Helmet Streamer**: See Streamer.
**High Frequency (HF)**: That portion of the radio frequency spectrum between 3 and 30 MHz.
**Heliospheric Plasma Sheet** (HPS): A high density slow solar wind region that is located

adjacent to the heliospheric current sheet (HCS).



**High Speed Solar Wind (HSS): A** solar wind with speeds of 750 to 800 km/s emanating from
solar coronal holes. The HSS is characterized by embedded, particularly large amplitude
Alfvén waves. At the edges of HSSs, the velocities can be less due to superradial
expansion effects.

**Instability**: When an electron or ion distribution is sufficiently anisotropic, it becomes unstable
(instability), generating plasma waves. The anisotropic distribution provides a source of
free energy for the instability. A simple analog is a stick, which if stood on end is
"unstable," but which if laid on its side is "stable." In this analog, gravity pulls on the
stick and provides a source of free energy when the stick is stood on end.

**Interplanetary Magnetic Field** (IMF, Parker spiral): The magnetic field carried with the solar
wind and twisted into an Archimedean spiral by the Sun's rotation.

**Interplanetary Medium**: The volume of space in the solar system that lies between the Sun and
the planets. The solar wind flows in the interplanetary medium.

**Interplanetary Coronal Mass Ejection (ICME)**: The evolutionary part of a CME as it propagates
through interplanetary space. Typically after the CME has propagated 1 AU from the Sun,
the ICME only contains the magnetic cloud (MC) portion of the initial three parts of a
CME. The MC may also have been compressed/expanded or rotated by the time it reaches
1 AU.

**Interplanetary Shock**: A fast forward shock is characterized by a sharp increase in solar wind
speed, plasma density, plasma temperature and magnetic field magnitude. The shock
reduces the upstream plasma from a supermagnetosonic state to a subsonic state, much as
an airplane wing sonic shock reduces the relative flow of air from a supersonic speed
(relative to the airplane) to a subsonic speed. A fast (magnetosonic) forward (propagating
in the direction of the "piston", in this case the propagation of the ICME in the antisolar
direction) shock is detected upstream (antisolarward) of fast ICMEs. A reverse shock
propagates in the direction of the Sun. Planetary bow shocks are reverse shocks. There
are other types of shocks not discussed in this paper: slow shocks and intermediate shocks.



**Interstellar** (gas, neutral gas, ions, cosmic rays, wind, magnetic field, etc.) Literally, between
the stars. In practical terms, it is anything beyond the outer boundary of the solar wind
(the "heliopause") yet within the Milky Way.
**Ion**: (1). An electrically charged atom or molecule. (2). An atom or molecular fragment that has
a positive electrical charge due to the loss of one or more electrons; the simplest ion is the
hydrogen nucleus, a single proton.
**Ionization State**: The number of electrons missing from an atom.
**Ionosphere**: The region of the Earth's upper atmosphere containing free (not bound to an atom
or molecule) electrons and ions. This ionization is produced from the neutral atmosphere
by solar ultraviolet radiation at very short wavelengths (<100 nm) and also by
precipitating energetic particles.
**Ionospheric Storm**: A positive ionospheric storm is where the ionospheric total electron content
(TEC) increases. A negative ionospheric storm is an event where the ionospheric TEC
decreases.
**Ionospheric Connection Explorer (ICON):** is a NASA 2-year mission that will give new views
of the boundary between our atmosphere and space, where planetary weather and space
weather meet.
**Irradiance**: Radiant energy flux density on a given surface (e. g. ergs $cm^{-2}s^{-1}$).
**keV**: 1000 electron Volts (kiloelectron Volt). See electron Volt. See also Anisotropic Plasma;
Plasma.
**Loop** (solar-loop prominence system): A magnetic loop is the flux tube which crosses from one
polarity to another. A loop prominence bridges a magnetic inversion line across which
the magnetic field changes direction. See also Magnetic Foot Point; Prominence.
**Loss Cone:** A small cone angle about the ambient magnetic field direction where
magnetospheric charged particles with velocity vectors within the cone will mirror at
sufficiently low altitudes such that the particle will have collisions with atmospheric
atoms and molecules and will be "lost" from returning to the magnetosphere.





**Loss Cone Instability**: An instability generated by a plasma anisotropy where the temperature perpendicular to the magnetic field is greater than the temperature parallel to the field. This instability gets its name because this condition exists in the Earth's magnetosphere and the "loss cone" particles are those that are lost into the upper atmosphere.

**Magnetic Cloud**: A region in the solar wind of about 0.25 AU or more in radial extent in which the magnetic field strength is high and the direction of one component of the magnetic field changes appreciably by means of a rotation nearly parallel to a plane. Magnetic clouds may be parts of the driver gases (coronal mass ejections) in the interplanetary medium.

**Magnetic Foot Point**: For the Earth's magnetic field lines, where the magnetic field enters the surface of the Earth.

**Magnetic Mirror**: Char particles moving into a region of converging magnetic flux (as at the pole of a magnet) will experience "Lorentz" forces that slow the particles and "mirror" them by eventually reversing their direction if the particles are initially moving slowly enough along the field line. See also Mirror Point.

**Magnetic Reconnection**: The act of interconnection between oppositely directed magnetic field lines.

**Magnetic Storm**: see Geomagnetic Storm.

**Magnetopause**: The boundary surface between the solar wind and magnetosphere, where the pressure of the magnetic field of the object effectively equals the ram pressure of the solar wind plasma.

**Magnetosheath**: The region between the bow shock and the magnetopause, characterized by very turbulent plasma. This plasma has been heated (shocked) and slowed as it passed through the bow shock. For the Earth, along the Sun-Earth axis, the magnetosheath is about 3 Earth radii thick.

**Magnetosonic Speed** (acoustic speed): The speed of the fastest low frequency magnetic waves in a magnetized plasma. It is the equivalent of the sound speed in a neutral gas or non-magnetized plasma.

**Magnetosphere**: The magnetic cavity surrounding a magnetized planet, carved out of the passing solar wind by virtue of the planetary magnetic field, which prevents, or at least impedes, the direct entry of the solar wind plasma into the cavity.

**Magnetospheric Multiscale Mission (MMS)**: A NASA mission designed to spend extensive periods in locations where magnetic reconnection at the magnetopause/magnetotail is expected to occur. The critical electron diffusion region will be studied. The mission consists of 4 spacecraft flown in a tetrahedron configuration.

**Magnetotail**: The extension of the magnetosphere in the anti-sunward direction as a result of interaction with the solar wind. In the inner magnetotail, the field lines maintain a roughly dipolar configuration. But at greater distances in the anti-sunward direction, the field lines are stretched into northern and southern lobes, separated by a plasmasheet. There is observational evidence for traces of the Earth's magnetotail as far as 1000 Earth radii downstream, in the anti-solar direction.

**Maxwellian Distribution**: The minimum energy particle distribution for a given temperature. It is also the equilibrium distribution in the absence of losses due to radiation, collisions, etc.

**Mean Free Path**: The statistically most probably distance a particle travels before undergoing a collision with another particle or interacting with a wave.

**Mesosphere**: The region of the Earth's atmosphere between the upper limit of the stratosphere (approximately 30 km altitude) and the lower limit of the thermosphere (approximately 80 km altitude).

**MeV**: One million electron Volts (Megaelectron Volt). See also Electron Volt.

**Mirror Point**: The point where the charged particles reverse direction (mirrors). At this point, all of the particle motion is perpendicular to the local ambient magnetic field. See also Magnetic Mirror.

**Parker Solar Probe**: a NASA robotic spacecraft to probe the outer corona of the Sun. It will approach to within 9.9 solar radii (6.9 million kilometers or 4.3 million miles from the

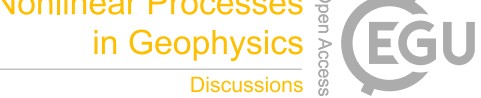

center of the Sun and will travel, at closest approach, as fast as 690,000 km/h

(430,000 mph).

**Photosphere**: The lowest visible layer of the solar atmosphere; corresponds to the solar surface

viewed in white light. Sunspots and faculae are observed in the photosphere.

**Pickup Ion**: An ion which has entered the solar system as a neutral particle and then becomes

ionized either through charge exchange or photoionization. It is called a pickup ion

because as soon as the neutral atom is ionized, it becomes attached to the magnetic field

carried by the solar wind and so is "picked up" by the solar wind.

**Pitch Angle**: In a plasma, the angle between the instantaneous velocity vector of a charged

particle and the direction of the ambient magnetic field.

**Plasma** (ions, electrons): A gas that is sufficiently ionized so as to affect its dynamical behavior.

A plasma is a good electrical conductor and is strongly affected by magnetic fields. See

also Anisotropic Plasma; Isotropic Plasma.

**Plasma Instability** (ion, electron): When a plasma is sufficiently anisotropic, plasma waves

grow, which in turn alter the distribution via wave-particle interactions. The plasma is

"unstable."

**Plasma Sheet**: A region in the center of the magnetotail between the north and south lobes. The

plasma sheet is characterized by hot, dense plasma and is a high beta plasma region, in

contrast to the low beta lobes. The plasma sheet bounds the neutral sheet where the

magnetic field direction reverses from Earthward (north lobe direction) to anti-Earthward

(south lobe direction).

**Plasma Wave** (electrostatic/electromagnetic): A wave generated by plasma instabilities or other

unstable modes of oscillation allowable in a plasma. "Chorus" and "Plasmasheric Hiss"

are whistler wave modes.  These are electromagnetic waves with frequencies below the

electron cyclotron frequency.  Electromagnetic ion cyclotron (EMIC) waves are ion

cyclotron waves with frequencies below the proton cyclotron frequency.

**Polar Cap Absorption Event (PCA)**: An anomalous condition of the polar ionosphere whereby

HF and VHF (3-300 MHz) radio waves are absorbed, and LF and VLF (3-300 kHz) radio



waves are reflected at lower altitudes than normal. The cause is energetic particle
precipitation into the ionosphere/atmosphere. The enhanced ionization caused by this
precipitation leads to cosmic radio noise absorption and attenuation of that noise at the
surface of the Earth. PCAs generally originate with major solar flares, beginning within a
few hours of the event (after the flare particles have propagated to the Earth) and
maximizing within a day or two after onset. As measured by a riometer (relative
ionospheric opacity meter), the PCA event threshold is 2 dB of absorption at 30 MHz for
daytime and 0.5 dB at night. In practice, the absorption is inferred from the proton flux at
energies greater than 10 MeV, so that PCAs and proton events are simultaneous.
However, the transpolar radio paths may still be disturbed for days, up to weeks,
following the end of a proton event, and there is some ambiguity about the operational
use of the term PCA.
**Prominence**: A term identifying cloud-like features in the solar atmosphere. The features appear
as bright structures in the corona above the solar limb and as dark filaments when seen
projected against the solar disk. Prominences are further classified by their shape (for
example, mound prominence, coronal rain) and activity. They are most clearly and most
often observed in H alpha. See also Loop.
**Radiation Belt**: Regions of the magnetosphere roughly 1.2 to 6 Earth radii above the equator in
which charged particles are stably trapped by closed geomagnetic field lines. There are
two belts. The inner belt's maximum proton density lies near 5000 km above the Earth's
surface. Inner belt protons (10s of MeV) and electrons (100s of keV) and originate from
the decay of secondary neutrons created during collisions between cosmic rays and upper
atmospheric particles. The outer belt extends on to the magnetopause on the sunward side
(10 Earth radii under normal quiet conditions) and to about 6 Earth radii on the nightside.
The altitude of maximum proton density is near 16,000-20,000 km. Outer belt protons
and electrons are lower energy (about 200 eV to 1 MeV). The origin of the particles
(before they are energized to these high energies) is a mixture of the solar wind and the
ionosphere. The outer belt is also characterized by highly variable fluxes of energetic
electrons. The radiation belts are often called the "Van Allen radiation belts" because





they were discovered in 1958 by a research group at the University of Iowa led by
Professor J. A. Van Allen. See also Trapped Particle.
**Ram Pressure**: Sometimes called "dynamic pressure". The pressure exerted by a streaming
plasma upon a blunt object.
**Reconnection**: A process by which differently directed field lines link up allowing topological
changes of the magnetic field to occur, determining patterns of plasma flow, and resulting
in conversion of magnetic energy to kinetic and thermal energy of the plasma.
Reconnection is invoked to explain the energization and acceleration of the
plasmas/energetic particles that are observed in solar flares, magnetic substorms and
storms, and elsewhere in the solar system.
**Relativistic**: Charged particles (ions or electrons) which have speeds comparable to the speed of
light.
**Ring Current**: In the magnetosphere, a region of current that flows near the geomagnetic
equator in the outer belt of the two Van Allen radiation belts. The current is produced by
the gradient and curvature drift of the trapped charged particles of energies of 10 to 300
keV. The ring current is greatly augmented during magnetic storms because of the hot
plasma injected from the magnetotail and upwelling oxygen ions from the ionosphere.
Further acceleration processes bring these ions and electrons up to ring current energies.
The ring current (which is a diamagtic current) causes a worldwide depression of the
horizontal geomagnetic field during a magnetic storm.
**Solar Energetic Particle (SEP)**: An energetic particle of solar flare/interplanetary shock origin.
**Sheath**: The plasma and magnetic fields in the downstream subsonic region after collisionless
shocks. See Shock Wave.
**Shock Wave**: A shock wave is characterized by a discontinuous change in pressure, density,
temperature, and particle streaming velocity, propagating through a compressible fluid or
plasma. Fast collisionless shock waves occur in the solar wind when fast solar wind
overtakes slow solar wind with the difference in speeds being greater than the
magnetosonic speed. Collisionless shock thicknesses are determined by the proton and



electron gyroradii rather than the collision lengths. See also Diffusive Shock

Acceleration; Solar Wind Shock.

**Solar Activity**: Transient perturbations of the solar atmosphere as measured by enhanced x-ray

emission (see x-ray flare class), typically associated with flares. Five standard terms are

used to describe the activity observed or expected within a 24-h period:

Very low - x-ray events less than C-class.

Low - C-class x-ray events.

Moderate - isolated (one to 4) M-class x-ray events.

High - several (5 or more) M-class x-ray events, or isolated (one to 4).

M5 or greater x-ray events.

Very high - several (5 or more) M5 or greater x-ray events.

**Solar Corona**: See Corona.

**Solar Cycle**: The approximately 11 year quasi-periodic variation in the sunspot number. The

polarity pattern of the magnetic field reverses with each cycle. Other solar phenomena,

such as the 10.7-cm solar radio emission, exhibit similar cyclical behavior. The solar

magnetic field reverses each sunspot cycle so there is a corresponding 22 year solar

magnetic cycle.

**Solar Energetic Particle (SEP) Event**: A high flux event of solar (low energy) cosmic rays.

This is commonly generated by larger solar flares or CME shocks, and lasts, typically

from minutes to days. See also Cosmic Ray.

**Solar Maximum**: The month(s) during the sunspot cycle when the smoothed sunspot number

reaches a maximum.

**Solar Minimum**: The month(s) during the sunspot cycle when the smoothed sunspot number

reaches a minimum.

**Solar Orbiter**: A European Space Agency-led (ESA) mission intended to perform detailed
measurements of the inner heliosphere and nascent solar wind to answer the question "How does





the Sun create and control the heliosphere?" The mission will make observations of the Sun from
an eccentric orbit moving as close as ~60 solar radii ($R_S$), or 0.284 astronomical units (AU) from
the Sun.

**Solar Wind**: The outward flow of solar particles and magnetic fields from the Sun. Typically at

1 AU, solar wind velocities are 300-800 km/s and proton and electron densities of 3-7 per

cubic centimeter (roughly inversely correlated with velocity). The total intensity of the

interplanetary magnetic field is nominally 3-8 nT.

**Space Weather**: Dynamic variations at the Sun, in interplanetary space, in the Earth's and

planetary magnetospheres, ionospheres and atmospheres associated with space

phenomena.

**Stratosphere**: That region of the Earth's atmosphere between the troposphere and the

mesosphere. It begins at an altitude of temperature minimum at approximately 13 km and

defines a layer of increasing temperature up to about 30 km.

**Streamer**: A feature of the white light solar corona (seen in eclipse or with a coronagraph) that

looks like a ray extending away from the Sun out to about 1 solar radius, having an arch-

like base containing a cavity usually occupied by a prominence.

**Substorm**: A substorm corresponds to an injection of charged particles from the magnetotail into

the nightside magnetosphere. Plasma instabilities lead to the precipitation of the particles

into the auroral zone ionosphere, producing intense aurorae. Potential drops along

magnetic field lines lead to the acceleration of ~1 to 10 keV electrons with brilliant

displays of aurora as the electrons impact the upper atmosphere.  Enhanced ionospheric

conductivity and externally imposed electric fields lead to the intensification of the

auroral electrojets.

**Sudden Impulse (SI):**  An abrupt (10s of seconds) jump in the Earth's surface magnetic field.

The positive sudden impulses (SI$^+$s) are caused by fast forward shock impingement onto

the magnetosphere.



**Sunspot**: An area seen as a dark spot, in contrast with its surroundings, on the photosphere of the Sun. Sunspots are concentrations of magnetic flux, typically occurring in bipolar clusters or groups. They appear dark because they are cooler than the surrounding photosphere. Larger and darker sunspots sometimes are surrounded (completely or partially) by penumbrae. The dark centers are umbrae. The smallest, immature spots are sometimes called pores.

**SWARM**: A European Space Agency (ESA) mission originally instrumented to study the Earth's magnetic field. The current goals are to study magnetospheric-ionospheric coupling and auroral space weather problems.

**Telsa**: A unit of magnetic flux density (Weber/m$^2$).  A nanoTesla (nT) is $10^{-9}$ Teslas.

**Termination Shock**: The shock wave in the solar wind which is caused by the abrupt deceleration of the solar wind as it runs into the local interstellar medium (LISM). It is thought to lie somewhere between 70 and 150 AU from the Sun.

**Thermal Speed** (ion, electron): The random velocity of a particle associated with its temperature.

**Thermosphere**: That region of the Earth's atmosphere where the neutral temperature increases with height. It begins above the mesosphere at about 80-85 km and extends upward to the exosphere.

**Total Electron Content (TEC)**: The column density of electrons in the Earth's ionosphere.

**Trapped Particle**: Particles gyrating about magnetic field lines (e.g., in the Earth's magnetosphere). See also Magnetic Mirror and Pitch Angle.

**Troposphere**: The lowest layer of the Earth's atmosphere, extending from the ground to the stratosphere, approximately 13 km altitude. In the troposphere, temperature decreases with height.

**Ultraviolet (UV)**: That part of the electromagnetic spectrum between 5 and 400 nm.

**Ultra Low Frequency (ULF)**: 1 milliHertz to 1 Hertz.



**Very High Frequency (VHF)**: That portion of the radio frequency spectrum from 3 to 300 MHz.

**Very Low Frequency (VLF)**: That portion of the radio frequency spectrum from 3 to 300 kHz.

**Van Allen Radiation Belt**: See Radiation Belt.

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

Acknowledgements.  Portions of this research were performed at the Jet Propulsion Laboratory,
California Institute of Technology under contract with NASA. GSL thanks the National Academy of
Sciences, India for support under the NASI-Senior Scientist Platinum Jubilee Fellowship Scheme. The
work of RH is funded by the Science & Engineering Research Board (SERB), a statutory body of
the Department of Science & Technology (DST), Government of India  through the Ramaujan
Fellowship.