# Peer review of "The Physics of Space Weather/Solar Terrestrial Physics (STP): What We Know Now and"

_Nonlinear Processes in Geophysics, 2019_

## Referee Comment (RC1) · Anonymous Referee #1 · 21 Aug 2019

The manuscript "Space Weather Forecasting: What we know now and what are the current and future challenges" submitted to NPG, 2019, by Tsurutani et al., represents overall an excellent summary of the physics background of geomagnetic storms and substorms, solar energetic particle fluxes, enhanced energetic magnetospheric electron fluxes and radiation belt formation, as well as ionospheric TEC changes. The focus is placed on the physics of the different space processes and the interplanetary causes and solar origins.

It is very understandable that a summary on space weather cannot easily cover all aspects (incl. the glossary), ranging from new solar observations from SDO, 3D CME

modelling results based on STEREO observations, new heliospheric imaging results for ICMEs also from STEREO, CME/ICME kinematics and new projects (e.g., FLARE-CAST, HESPERIA, AFFECTS, HELCATS, etc. and also other US and int. projects). However, before publication I suggest to state this in the paper and to add ref. about ongoing projects and literature covering those issues. I am not pointing out special references because they are easy to find through the internet or by browsing the Space Weather Journals of AGU and Int. Journal on Space Weather and Climate. I suggest adding clarifications of the focus and limitations of the paper at the beginning and end of the manuscript, see also the specific comments below. I also suggest to name some books on space weather (e.g., Hanslmeier: The Sun and space weather; Koskinen: The physics of space storms, etc.). Eventually even the title maybe modified to these suggestions to be more specific.

I further suggest adding more specific details on how new missions (PSP, SO) will help answer the addressed questions, or if it is not possible, to leave it out.

With these modifications the paper will certainly be a very good overview on space weather processes, written in a clear way.

Minor comments:

1) p.1, l.11: Since also the solar wind speed plays a role because of E=-v x Bz, I suggest to add the word "major" at the beginning, i.e., "Major geomagnetic storms are caused by ...". Or some similar clarification.

2) Same p., l.17: I suggest adding a sentence on SEPs because the topic start a little abrupt.

3) p.2, l.35: I suggest removing the word "old" by a more elegant sentence stating an evolution of space weather from solar terrestrial research over the years, or something similar. Several space missions over the last decades have made significant progress in terms of interdisciplinary research (SOHO, Cluster, ACE, STEREO, etc.) between

the solar, magnetospheric, ionospheric disciplines and space physics in general. And the new data have led to fundamental new insights into solar storms (e.g., CME 3D structure and propagation to Earth).

4) p. 3, l. 70: I suggest adding "that occur more frequent during . . .".

5) Same p., next lines: I suggest to rephrase " We will explain to solar scientists . . .". There are also solar scientists knowledgeable of space physics.

6) p.6, 1st par.: I suggest to not completely neglect the role of v solar wind here. I see it is addressed later on.

7) P.7, l.180: lASCO has observed by now far more then 10.000 CMEs, but only about 5% are faster than 700 km/s in the plane of sky. Only a very few have speeds of >2.000 km/s and these are coming preferentiall from coronal regions above enhanced photospheric fields, so that higher field strengths and compression effects are pronounced. That means only a subset of CMEs produces strong fields in ip space. Please add some clarifications.

8) p.8, 2nd par.: Results from STEREO observations are missing here. It is also pointed out that new missions will provide new insights, but do they really do for these research topics? And if so, how?

9) p.11, l. 268-272: There are results that relate MC magnetic field structures back to their solar source regions. I suggest including a few sentences.

10) Same p., next par.: Again, how do PSP and SO help specifically?

11) p.17, l.414: Only "intensities that some MC fields do". Many MCs have weaker fields.

12) Same p., l.422: "having said"?

13) p.18, caption Fig.7: I suggest to write: "A large coronal hole at the . . ."

14) p.19, l.442-444: LASCO C2 is also included.

15) p.21, l.480: How will these missions be useful? Be specific.

16) p.23, l.533: same as 15).

17) p.32, l.724: "stronger"

18) same p., l.727: wording of sentence

19) p.35, l.805-812: I missed some results from STEREO in this context.

20) Same p., 816-824: I suggest including here some sentences on the established drag modelling for CME propagation I the heliosphere.

21) p.37, l.865: "have shown"

22) p.38, l. 887: wording of sentence

23) p.39, caption Fig.18: I suggest adding clarifying text about the shock creation.

24) p.42, l.962: I suggest removing the word "poor". Either there is connection or not.

25) p.50, l.1100: Why is the magnetic profile unlike those of other ICMEs? Please explain.

26) p.53, caption Fig.29: Please add the date.

27) p.54, l.1196: Please explain what the averaging time for Bz was to avoid averaging out negative time intervals.

28) p.55, l. 1225: What is meant by a solar filament in this context? Please explain.

29) p.31, caption Fig. 31: Please add the year.

30) p.57, l.1257-1260: I suggest elaborating things not addressed a little bit, see major comments.

31) Same p., next par.: I suggest adding the role of V, also for CME arrival time predictions.

32) p.58, first 2 lines: But what about forecasting with ENLIL?

33) Same p., last par.: I suggest adding drag modelling here.

34) p. 59, last lines: I suggest to add some more concluding remarks and references to books on space weather, including recently established forecast models and new projects.

35) p.60, l. 1338: wording of sentence

36) p.67, l. 1530: I suggest adding a statement on GNSS.

37) P.75, l. 1746: Solar activity includes many other phenomena, e.g., CMEs, jets, etc. but here only flares are addressed.

---

## Referee Comment (RC2) · Anonymous Referee #2 · 2 Sep 2019

[Main comments]

In this review paper, the authors summarized observations of space weather phenomena and their physical interpretations. The main topic is about geomagnetic storms and magnetospheric phenomena, which is based on the authors' previous studies, ranging from the arrival of ICMEs and solar wind plasma at the Earth to the resulting geomagnetic and ionospheric storms. Phenomenological understanding is broadly explained, and questions about unresolved problems are described in each section, leading to what to reveal by new space missions like PSP, Solar Orbiter, MMS, Arase and SWARM.

[Figure]

This paper is written not only for space plasma physicists but for non-space plasma readers, like solar physicists and ionospheric scientists. It looks that the authors hope lots of people to read this article and try the unresolved problems with the interdisciplinary cooperation. The terminologies are summarized at the end of the main text, and in each section, histories of the studies are explained, which are useful for beginners and young researchers.

On the other hand, though the title is "Space Weather Forecasting", the manuscript does not cover predictions of solar flares, CMEs, SEPs, GICs and plasma bubbles, as well as social impacts on the infrastructures. The methods using numerical simulations and machine-learning techniques are not well introduced in this paper. It would be more useful for readers if the authors can include the current prediction models and their prediction accuracy in this review paper.

Especially, with a new approach using machine-learning algorithms, probabilistic predictions can be done even if the physical mechanisms are not fully revealed. For accurate forecasting, the full understanding of physical processes are really necessary? If we understand all the nonlinear processes in space weather phenomena, can we forecast them perfectly? It would be also useful for readers if the authors can answer these questions.

[Minor comments]

1) [Fig. 7] It looks that the solar image is not from SDO but Yohkoh. SDO does not have a soft X-ray telescope.

2) [Fig. 8] The inner solar image was not taken by a soft X-ray telescope but an EUV telescope of EIT (195A Fe XII). The inner coronal image in the black circle was taken by Mauna Loa coronagraph, while the outer one was by SOHO/LASCO-C2.

3) [section 2.4.1] There is a sentence that to determine IMF-Bz component in the sheaths, we need more effort on predicting the slow solar wind plasma and magnetic
field, but this statement is obscure. What will be the key to predict the slow solar wind plasma?

4) [General comments] There are so many abbreviations like MC, ICME, IMF, CIR, HSS, HCS, HPS, HILDCAA, AE, EIA, EMIC wave, PC wave, RED, PPEF, SSW, SSS, which are difficult for non-space plasma readers to understand.

5) It's better to show the definition of L value.

---

## Author Comment (AC1) · 15 Nov 2019

The manuscript "Space Weather Forecasting: What we know now and what are the current and future challenges" submitted to NPG, 2019, by Tsurutani et al., represents overall an excellent summary of the physics background of geomagnetic storms and substorms, solar energetic particle fluxes, enhanced energetic magnetospheric electron fluxes and radiation belt formation, as well as ionospheric TEC changes. The focus is placed on the physics of the different space processes and the interplanetary causes and solar origins.

It is very understandable that a summary on space weather cannot easily cover all aspects (incl. the glossary), ranging from new solar observations from SDO, 3D CME modelling results based on STEREO observations, new heliospheric imaging results for ICMEs also from STEREO, CME/ICME kinematics and new projects (e.g., FLARECAST, HESPERIA, AFFECTS, HELCATS, etc. and also other US and int. projects). However, before publication I suggest to state this in the paper and to add ref. about ongoing projects and literature covering those issues. I am not pointing out special references because they are easy to find through the internet or by browsing the Space Weather Journals of AGU and Int. Journal on Space Weather and Climate. I suggest adding clarifications of the focus and limitations of the paper at the beginning and end of the manuscript, see also the specific comments below. I also suggest to name some books on space weather (e.g., Hanslmeier: The Sun and space weather; Koskinen: The physics of space storms, etc.). Eventually even the title maybe modified to these suggestions to be more specific.

I further suggest adding more specific details on how new missions (PSP, SO) will help answer the addressed questions, or if it is not possible, to leave it out.

With these modifications the paper will certainly be a very good overview on space weather processes, written in a clear way.

We thank the referee for his/her helpful comments. Based on your comments and that of the other referee, we now reduce the usage of the word "forecasting" and speak mainly of the physics of space weather, a point that you have emphasized. This is now in the title of the paper. This was our original intent. We thank you for the references to the Hanslmeier and Koskinen books. We have mentioned those, some earlier ones and one more very recent publication (Buzulukova, 2018) in the Introduction section and state that Space Weather is an extremely broad field and that even those many books have not covered all areas of importance. Our present effort is not only to fill in the cracks but to give a different slant to the topic of Space Weather and what fruitful research can be done today and in the next 10 to 25 years. This is now explicitly stated in the paper.

We now give more specific ideas on how current and future missions could help solve outstanding questions.

Minor comments:

1) p.1, l.11: Since also the solar wind speed plays a role because of E=-v x Bz, I suggest to add the word "major" at the beginning, i.e., "Major geomagnetic storms are caused by ...". Or some similar clarification.

Yes, Corrected.

2) Same p., l.17: I suggest adding a sentence on SEPs because the topic start a little abrupt.

Done. A phrase was added.

3) p.2, l.35: I suggest removing the word "old" by a more elegant sentence stating an evolution of space weather from solar terrestrial research over the years, or something similar. Several space missions over the last decades have made significant progress in terms of interdisciplinary research (SOHO, Cluster, ACE, STEREO, etc.) between the solar, magnetospheric, ionospheric disciplines and space physics in general. And the new data have led to fundamental new insights into solar storms (e.g., CME 3D structure and propagation to Earth).
* * *
Okay. Done. The main point that we wanted to make to the reader is that many of the physical phenomena have already been discovered and much of the future work is fine tuning, and understanding of the detailed physics.

4) p. 3, l. 70: I suggest adding "that occur more frequent during ...".

Corrected. Thank you.

5) Same p., next lines: I suggest to rephrase " We will explain to solar scientists ...". There are also solar scientists knowledgeable of space physics.

Yes, corrected. We now address this "to the reader". Sorry. This phrase came when one of us gave a talk on space weather at a meeting and afterwards he was collared by a very prominent solar physicist who thanked the speaker for defining why space weather people talk about ICMEs. This person in the audience did not know.

6) p.6, 1st par.: I suggest to not completely neglect the role of v solar wind here. I see it is addressed later on.

Yes, okay. Done. It should be mentioned to the referee that the variation in V typically ranges from ~400 km/s to 1,000 km/s, a factor of about 2. However the Bsouth component varies from

about 0 nT to say -60 nT for a major magnetic storm.  The variation in Bsouth is the greater of the two.

> 7) P.7, l.180: lASCO has observed by now far more then 10.000 CMEs, but only about 5% are faster than 700 km/s in the plane of sky. Only a very few have speeds of >2.000 km/s and these are coming preferentially from coronal regions above enhanced photospheric fields, so that higher field strengths and compression effects are pronounced. That means only a subset of CMEs produces strong fields in ip space. Please add some clarifications.

Thank you very much for the information. We have paraphrased your comments in quotation marks and have added it to the paper.  We did not know this.  Very important.

> 8) p.8, 2nd par.: Results from STEREO observations are missing here. It is also pointed out that new missions will provide new insights, but do they really do for these research topics? And if so, how?

Corrected. Thank you.  It is obvious that STEREO should have addressed this issue already (but haven't).  In our initial writeup we were only focusing on future missions.  A reference to STEREO has been added.

> 9) p.11, l. 268-272: There are results that relate MC magnetic field structures back to their solar source regions. I suggest including a few sentences.

Corrected.

> 10) Same p., next par.: Again, how do PSP and SO help specifically?

Corrected.  We now mention that they would have to study the same ICME at different distances from the Sun.

> 11) p.17, l.414: Only "intensities that some MC fields do". Many MCs have weaker fields.

Corrected.

> 12) Same p., l.422: "having said"?

Corrected.

> 13) p.18, caption Fig.7: I suggest to write: "A large coronal hole at the ..."

Thank you.  Corrected.

> 14) p.19, l.442-444: LASCO C2 is also included.

Now corrected.

> 15) p.21, l.480: How will these missions be useful? Be specific.

Amplified and corrected.

16) p.23, l.533: same as 15).

Corrected.

17) p.32, l.724: "stronger"

Corrected.

18) same p., l.727: wording of sentence

Corrected.

19) p.35, l.805-812: I missed some results from STEREO in this context.

Corrected.

20) Same p., 816-824: I suggest including here some sentences on the established drag modelling for CME propagation I the heliosphere.

Corrected.

21) p.37, l.865: "have shown"

Corrected.

22) p.38, l. 887: wording of sentence

Corrected.

23) p.39, caption Fig.18: I suggest adding clarifying text about the shock creation.

Corrected. We have added more explanation of the figure in both the figure caption and within the text.

24) p.42, l.962: I suggest removing the word "poor". Either there is connection or not.

Corrected.

25) p.50, l.1100: Why is the magnetic profile unlike those of other ICMEs? Please explain.

Corrected. We now mention that the profile is discussed later in the paper. We have expanded that discussion (later in the paper).

26) p.53, caption Fig.29: Please add the date.

Done.

27) p.54, l.1196: Please explain what the averaging time for Bz was to avoid averaging out negative time intervals.

Done. We have also added a comment on magnetospheric reaction timescales.

28) p.55, l. 1225: What is meant by a solar filament in this context? Please explain.

Corrected. This filament is the interplanetary manifestation of the Illing and Hundhausen (1986) CME filament. We now mention this in the text.

29) p.31, caption Fig. 31: Please add the year.

Done.

30) p.57, l.1257-1260: I suggest elaborating things not addressed a little bit, see major comments.

Amended. The details follow.

31) Same p., next par.: I suggest adding the role of V, also for CME arrival time predicC4 tions.

Yes. Okay. Very important. Done.

32) p.58, first 2 lines: But what about forecasting with ENLIL?

To address the issue of using physics based computer codes, data based computer codes and machine learning algorithms, we have made some "Final Comments" at the end of the paper. We have added references to many ENLIL based codes but remark that different codes have different strengths in predicting plasma properties and timing, but they do not address the MC magnetic field direction and intensity, the prime point of this paper. The sheath fields are also not well specified nor tested.

33) Same p., last par.: I suggest adding drag modelling here.

Some references had been added in the Final Comments. Basically one of the ENLIL references states that one has to have imaging data to do real-time studies of deceleration and acceleration. We think the eventual goals would be to model the slow solar wind upstream of the CME so that the slow solar wind can be modeled. This has not been done to our knowledge.

34) p. 59, last lines: I suggest to add some more concluding remarks and references to books on space weather, including recently established forecast models and new projects.

Done. We have added a "Final Comments" section at the very end of the paper. Rather than address details on forecasting models and new space projects, we thought it more important to emphasize model predictions of MC fields at 1 AU and testing which has not been done yet. Also many of the space projects typically do not address the fundamental physical problems mentioned here.

35) p.60, l. 1338: wording of sentence

Thank you. Done.

36) p.67, l. 1530: I suggest adding a statement on GNSS. 37) P.75, l. 1746: Solar activity includes many other phenomena, e.g., CMEs, jets, etc. but here only flares are addressed.

Corrected.  We changed the title of 'Solar activity" to "Solar Flares". CMEs are discussed separately.  We have added a section on GNSS.

---

## Author Comment (AC2) · 15 Nov 2019

[Main comments]

In this review paper, the authors summarized observations of space weather phenomena and their physical interpretations. The main topic is about geomagnetic storms and magnetospheric phenomena, which is based on the authors' previous studies, ranging from the arrival of ICMEs and solar wind plasma at the Earth to the resulting geomagnetic and ionospheric storms. Phenomenological understanding is broadly explained, and questions about unresolved problems are described in each section, leading to what to reveal by new space missions like PSP, Solar Orbiter, MMS, Araseand SWARM.

This paper is written not only for space plasma physicists but for non-space plasma readers, like solar physicists and ionospheric scientists. It looks that the authors hope lots of people to read this article and try the unresolved problems with the interdisciplinary cooperation. The terminologies are summarized at the end of the main text, and in each section, histories of the studies are explained, which are useful for beginners and young researchers.

On the other hand, though the title is "Space Weather Forecasting", the manuscript does not cover predictions of solar flares, CMEs, SEPs, GICs and plasma bubbles , as well as social impacts on the infrastructures. The methods using numerical simulations and machine-learning techniques are not well introduced in this paper. It would be more useful for readers if the authors can include the current prediction models and their prediction accuracy in this review paper.

Especially, with a new approach using machine-learning algorithms, probabilistic pre-dictions can be done even if the physical mechanisms are not fully revealed. For accurate forecasting, the full understanding of physical processes are really necessary? If we understand all the nonlinear processes in space weather phenomena, can we forecast them perfectly? It would be also useful for readers if the authors can answer these questions.

We thank the referee for his/her helpful comments in improving this paper. The points that you have raised above are very pertinent and are indeed topics that have not been covered well.  To address your specific comment about discussing predictions of solar, interplanetary, magnetospheric and ionospheric phenomena, we have decided to change the tone of the paper to indicate that we are addressing only the knowledge of the physical causes of such phenomena.  Our original thought was that we need to know the physical causes before making forecasts/predictions.  However the words "forecast/predictions" mean other things to other people (see comments from referee #1) and this can be confusing. We have therefore changed the title of the paper and part of the text to reflect this.

To address some of your other questions about "forecasting" using computer codes and machine learning algorithms, we address it here in detail in a Final Comments section at the end of the paper.  Many physics-based codes have been constructed over tens of years.

However most of them have not been tested even using data from past magnetic storms. People has simply assumed that with all the major "physics" put into the codes, that one would be able to predict observations. One of us (BTT) has been involved with a NASA funded project to test the codes for ionospheric total electron content (TEC) data. We have been at this for the last 5 years. We have been using the well-established codes with measured solar wind input and have had the CCMC personnel of the Goddard Space Flight Center run the codes for us. The results have not been good. Basically we are not able to get accurate reproductions of the observations from any of the codes. At this time, we have no idea why we are not getting the predicted results, especially with solar wind, solar and geomagnetic activity index inputs. We have submitted two papers on this topic and as one would expect, the referees are not happy. Well, we are not happy either, but we simply want to report our results so improvements can be made. So to answer one of your questions, the independent reporting on the accuracy of codes in predicting so far is essentially non-existent.

The other referee has mentioned the ENLIL code and related codes. We now reference many of these works. People have tested MC propagation from the Sun to 1 AU but only the solar wind plasma properties and arrival times. The MC itself has not been modelled and tested. This is now stated in the paper.

Concerning machine learning algorithms, that too can be a red herring. Some of us have been studying such applications. Although great claims of success by proponents have been made, actual space weather successes are rare to none. What if the ionosphere is dominated by chaos? Then machine learning will not help. We have added several references to this topic in the Final Comments section (a new section at the end of the paper).

From the above one can see that neither we nor anyone else is really qualified to talk about "forecasting" using computer codes or machine learning algorithms. There have been no tests for the topics addressed in this paper to the knowledge of the authors. On the other hand, we do not wish to put such future studies in a negative light. Perhaps someone will be able to make a verifiable breakthrough. We wish to be positive on this subject. To address the topic of "forecasting", we have made some short comments near the end of the paper.

We are very interested in what atmospheric weather people do and have been following their work closely. They have been diligently working at their problem far longer than space weather people have been. It is interesting to know how they make their predictions. They have many codes at their disposal. They down-select to say the ~25 best ones and then take the mean value! This seems to work well. But why it works leaves a big question? Maybe that is the answer for space weather as well.

We have passed our section of "Final Comments" to the other NASA funded JPL researchers (Xing Meng, Tony Mannucci (P.I.) and Olga Verkhyagladova). They are in agreement with the wording of this section. As mentioned before, we have been examining existing ionospheric TEC codes for the last 5 years.

[Minor comments]

1) [Fig. 7] It looks that the solar image is not from SDO but Yohkoh.  SDO does not have a soft X-ray telescope.

We apologize for the error!  Thank you very much for the correction.  This has been fixed.

2) [Fig.  8] The inner solar image was not taken by a soft X-ray telescope but an EUV telescope of EIT (195A Fe XII). The inner coronal image in the black circle was taken by Mauna Loa coronagraph, while the outer one was by SOHO/LASCO-C2.

Corrected.  Thank you.  We did not mention the outer coronagraph image previously.  We do now.

3) [section 2.4.1] There is a sentence that to determine IMF-Bz component in the sheaths, we need more effort on predicting the slow solar wind plasma and magnetic field, but this statement is obscure. What will be the key to predict the slow solar wind plasma?

Corrected.  Thank you.  Right now we have no idea on how to determine the properties of the slow solar wind, but as you note this is key to predicting the IMF Bz in sheaths.  We have reemphasized this point near the end of the paper.

4) [General comments] There are so many abbreviations like MC, ICME, IMF, CIR,HSS, HCS, HPS, HILDCAA, AE, EIA, EMIC wave, PC wave, RED, PPEF, SSW, SSS, which are difficult for non-space plasma readers to understand.

Yes, we have no solution to this problem other than to put in a Glossary so the reader can go back and forth.  Putting in the full spelling of the acronyms will lengthen the paper by a lot.

5) It's better to show the definition of L value.

Corrected.  We have inserted a definition in the text and also added this to the Glossary.

---

## Author Response (AR2)

Major comments:

We thank the authors' great effort to improve their manuscript, by taking account of the referees' comments. It became clear that the authors focus on the physical causes of phenomena, rather than forecasts in this paper. They changed the title, and it became more suitable to the topic of this paper. However, if the authors focus on physics, it may be more suitable to use "Solar-Terrestrial Physics" (STP) for the title.

Thank you. We have added this to the title. Actually we think Solar-Terrestial Physics is the same as Space Weather since all things in solar terrestrial phenomena are interconnected and can influence space weather. We mention this now in the paper. At least this is true for interplanetary, magnetospheric and ionospheric phenomena. When we used the term "forecasts" we were meaning in the sense that one understands the physics well enough to predict events in the future. We have now used this term sparingly because we realize that people have different thoughts about what this means.

I agree with the authors' idea that the application of weather forecasting will help us improve the predictability of space weather. In the point of view of space weather forecasts, it is more important to show how much we can predict phenomena even if they are chaotic or which phenomena can be predicted. Recently, machine-learning techniques have been applied to solar flare predictions, and they succeeded in improving prediction accuracy a lot.

We are very happy to hear these positive results for machine learning techniques applied to solar flare predictions. Some of us have reviewed papers on machine learning associated results. The authors often exaggerate their successes. Of course we remain positive about this new technique. All we are trying to say is others should objectively assess the successes and failures. One should not assume that this new technique will be a panacea for the problem.

I also agree that prediction models should be evaluated by the third parties, but in the same operational settings. Recently, operational models have been compared in CCMC/NASA and other small benchmark workshops. I feel that there is no standard evaluation method discussed in our community and it is a problem.

We understand your comment. We have added a few sentences in the paper addressing this issue. All of our new additions to the paper are written in blue (as is this response).

"The best test for proving that workers in Space Weather understand all of the underlying physics and/or the machine learning algorithm is robust is to use the program on a new event and see how well it does. This should be done by independent researchers like the people at CCMC at the Goddard Space Research Center, Greenbelt Maryland and/or other related facilities."

I read through the revised parts in the text, colored in blue. I have additional comments on them, and I think that the current manuscript is still not enough for publication in NPG. I hope that my comments are useful for the authors to improve their manuscript.

Detailed comments:

1) [p.2, section 1.1, 1st para., line 1] The sentence "Space Weather is a new term for a topic that actually began over a century and a half ago." is confusing. The term "space weather forecasting" first appeared in 1988 (Nagai 1988 JGR, Marubashi 1989 SSRv), and US national space weather program started in 1993. Please modify the sentence for readers to avoid misunderstanding.

Corrected as best we could. Please notice that we did not use the word "forecasting " as you have mentioned above. This particular word has different meanings to different people and we have therefore avoided the useage here. We now say "Space Weather is a new term for a topic/science that began over a century and a half ago". We hope that this is clearer now. Notice that after this statement we give concrete examples of Carrington, Maunder, Chree, Chapman, Bartels, Newton, etc. works that were published well before 1988. These works were all Space Weather science articles. We think the term "space weather" is just a new term describing of an old science.

Many of the young scientists don't understand that there were major space weather works published well before they were born. They probably don't remember or cite the papers that you mentioned. The people in the US think that the term was invented in the US. However the first coinage of the word is not so important. But it is important that young people realize that there were major contributions made to space weather science a long time ago.

2) [p.9, 2nd para, line 3-4] In the sentence "Only very few CMEs have speeds >2,000 km/s... (personal communications with referee, 2019)", it is better to cite some references (e.g. Yashiro et al.).
* * *
Done. Thank you.

3) [p.16, 2nd para, line 3-4] Is it correctly "y is in the direction of the Earth's south magnetic pole, (Omega x X)/|Omega x X|, where Omega is the angular velocity"?

Corrected. We have checked the statement in the paper and it is correct as it stands. We have added a phrase to help clarify things. To answer your question specifically, the x axis points towards the Sun. Omega is the direction of the south magnetic pole (which is close to the direction of the north rotational pole). So in the right hand system z = x x y. Z is in a direction close to the north rotational pole.

4) [p.20, section 3.2.1, 1st para. & Fig. 7, caption] Correctly, "the Soft X-ray Telescope (SXT)" onboard Yohkoh.

Corrected. Thank you.

5) [p.47, 1st para, line 2 and Fig. 22] It is hard to see the characters in Fig. 22 (c)-(d). Please cite the reference of this figure in the caption.

Done.

6) [p.47, 1st para, the last two sentences] It is better to remove the sentence "this solar flare example is... no transfer of energy to interplanetary space and then to the

magnetosphere.", because there is no evidence to show no energy transfer to the interplanetary space and the magnetosphere.

Okay, corrected. We have deleted the second sentence and have replaced it by another. The reason for stating this is to make magnetospheric scientists realize that not everything in space weather is transferred from the Sun to the Earth through ICMEs or high speed streams. That was the main intent of the sentence.

7) [p.47, 2nd para, line 1-2] The sentence "Some future space weather problems are to understand if the solar flare photon energy spectrum varies often" is misleading. Previous observations have shown that the energy spectra of solar flares differ in time and events. It is due to particle acceleration in flares, and the problem has been challenged by lots of solar physicists. I also recommend to use the term "solar energy spectrum" rather than "photon spectrum".

Okay, corrected. This has been reworded. One of us put this into the paper as a reaction to a solar physicist who strongly said at a meeting in reaction to a talk (about 10 years ago) "the solar flare spectrum does not vary!". The person's name is H. Hudson.

Actually this same person reacted somewhat violently to our review talk probably primarily because we were talking a little about solar physics (his topic and not ours). But in hindsight we thought then (and still do now), that there was nothing incorrect in what we said.

We apologize for reacting too strongly to a misstatement given by a solar physicist. This has been corrected.

We now mention that the challenge for solar physicists will be to "predict" the spectrum given the conditions in proximity to the flare site. This of course may be very difficult to do, but this is the theme of the whole paper: challenges for the future. Our point is if one can "predict it" you may understand the underlying physics. This is what we mean by the words "predict" or "forecast".

By the way we have been influenced by another famous solar physicist. After one talk when one of us explained the useage of the term Interplanetary CME, she came up to the speaker and thanked him for explaining the term. She had no idea why we were using this. This person was Shadia Habal. Again we have gone to some lengths to explain this term. We figured that if some solar people did not understand it, the person in the street would not at all. Thus our detailed explanation.

8) [p.48, line 1-2] The sentence "Solar flare data... would be useful to understand the details of flare spectral differences but solar physicists need to explain what the causes are" is also strange to solar physicists. The mechanisms of thermal and nonthermal emissions are qualitatively understood. For space weather forecasting, it is more important to predict X-ray/UV emissions from solar flares quantitatively. In addition, the solar energy spectrum is not observed by SORCE and TIMED, but by RHESSI and EVE/SDO.

Corrected. Thank you.

9) [p.62, Final comments, 1st para, line 7-10] "Meng et al. (2019b) have tested ...indices were used." In this sentence, "Meng et al." is not accepted now. What are the names of the two well-known ionospheric codes to predict TEC? The representation "not so good" is subjective, and the evaluation method is not explained at all. It is better to remove this sentence.

Deleted. These sentences have been removed.

The Meng et al. paper has just been resubmitted a third time will all questions addressed adequately. It should be accepted soon. We had looked for a quote in the paper for what "not so good" meant but there was none. The reason why this is not well stated in the paper is because the results were somewhat of a shock to the referees (and the authors).

10) [p.63, 3rd para] Who are the "atmospheric weather forecasters"? NOAA? Please cite references. As for the idea of ensemble forecast, it has already been included in machine-learning algorithms. So, it is useful mainly for scientists using models of physics-based numerical simulations.

Thank you. Corrected. We have added some references.

Concerning machine learning algorithms, we are saying something different. Even if different algorithms train on the same data and use ensemble forecasting they may get different results. Thus ~25 of these codes could be used and the results averaged. We did not change the wording for this part of the paper.

**Editor Decision: Publish subject to minor revisions (review by editor)** (29 Nov 2019) by Giovanni Lapenta

**Comments to the Author**:

Dear Dr. Tsurutani
the referee has still some significant comment. But once you address these final comments, the paper will be accepted.

Best regards

Giovanni

Dear Giovanni,

We have addressed all of the referee's comments in detail. We hope that the paper is now accepted for publication in NPG.

Bruce, Gurbax and Rajkumar